# Life Cycle Assessment for Transportation Infrastructure Policy Evaluation and Procurement for State and Local Governments

**John T. Harvey [1,*], Ali A. Butt [1,*], Mark T. Lozano [2], Alissa Kendall [3], Arash Saboori [1], Jeremy D. Lea [1], Changmo Kim [1] and Imad Basheer [4]**

[1] University of California Pavement Research Center, Department of Civil and Environmental Engineering, University of California, Davis, One Shields Avenue, Davis, CA 95616, USA; asaboori@ucdavis.edu (A.S.); jdlea@ucdavis.edu (J.D.L.); chkim@ucdavis.edu (C.K.)

[2] Energy Systems, Energy and Efficiency Institute, University of California, Davis, One Shields Avenue, Davis, CA 95616, USA; mtlozano@ucdavis.edu

[3] Department of Civil and Environmental Engineering and Energy and Efficiency Institute, University of California, Davis, One Shields Avenue, Davis, CA 95616, USA; amkendall@ucdavis.edu

[4] Pavement Program, California Department of Transportation, 2389 Gateway Oaks Drive, Sacramento, CA 95833, USA; imad.basheer@dot.ca.gov

* Correspondence: jtharvey@ucdavis.edu (J.T.H.); aabutt@ucdavis.edu (A.A.B.)

**Abstract:** Climate change is one of the defining challenges of our time, and achieving mitigation targets requires urgent action to identify and implement strategies for reducing greenhouse gas (GHG) emissions. However, identifying, quantifying, and then selecting among the many possible strategies to achieve GHG reductions is difficult, especially without a standardized approach for comparison. Presenting alternatives in a mitigation supply curve is an approach that has been used previously to compare the costs and magnitude of mitigation potential for different strategies. Some of the critiques of this approach include the lack of a consequential perspective in determining mitigation and the lack of a life cycle perspective in quantifying mitigation and economic costs. This research uses the principles of consequential life cycle assessment and life cycle cost analysis to improve on the mitigation supply curve concept to support evaluation and procurement decisions for transportation infrastructure. Results from pilot studies for road infrastructure indicate that a consequential life cycle approach for mitigation supply curves is feasible and can support agency decision-making and communication regarding those decisions.

**Keywords:** greenhouse gas emissions; life cycle assessment; life cycle cost analysis; supply curve; benefit–cost; conceptual analysis; transportation infrastructure

---

## 1. Introduction

Climate change has been identified as a likely existential threat to human civilization and ecological systems over a time period of decades [1,2]. Climate change pollutants, dominated by $CO_2$ from combustion of fossil fuels, is often emitted alongside criteria air pollutants that have direct and immediate effects on human health including reduced life expectancy and childhood respiratory problems, to name but a few [3–8]. Global, national, and regional governments have developed targets for mitigating greenhouse gases (GHGs) to reduce or avert climate change impacts [9–15]. GHG emissions and mitigation targets are often distinguished by the broad economic sector to which they belong, including industry, transportation, and residential/commercial buildings. Transportation, the large majority of which is combustion of fossil fuels by on-road vehicles, produces more than

40 percent of GHG emissions in California [16], though transportation's contribution varies by state and country, and is a lower percentage in most other states and countries [17].

The United States has over 4 million miles of public roads ([18] and 2.65 million miles of paved roadways [19], supporting nearly 3 trillion vehicle miles traveled annually [20]. The nation's roadway system is one part of a transportation network that provides mobility and access to a range of users (e.g., access to schools, services, and work; leisure travel; and general mobility) [21]. The roadway system is also vital to the economy because it enables the movement of freight and commodities and is a major source of employment. Roads carry about 65% of all freight in the nation, in terms of both tons and dollar value [22]. More than 300,000 people were employed in the road and bridge industry in 2014, and even more were employed before recent cuts in national transportation infrastructure funding. Most of these jobs do not require a college degree and typically offer higher wages than jobs requiring similar educational backgrounds [23].

However, operation of the nation's pavement network, which includes both its construction and its maintenance, is costly. The total annual construction and maintenance expenditure for U.S. highways (pavements and bridges) in 2014 was $165 billion [24]). Highway construction and maintenance also requires large inputs of energy and natural resources, and causes significant emissions of GHGs, criteria air pollutants, and water pollutants. Vehicle operation on the nation's roadways consumes more than 181 billion gallons of fuel [25] and the amount of energy consumed by vehicles is affected by the pavements they roll on.

Taken together, these numbers demonstrate the magnitude of the investment in public roadways, and the system's importance in supporting movement, access, and mobility. At the same time, there is increased recognition of the harm caused by pollutants from roadway construction and demolition of worn-out materials; of the influence of pavement on the fuel use of vehicles and on the surrounding environment. The transportation sector of the US economy produces about 28.5 percent of national GHG emissions [26].

California has established a series of mandated targets for reducing GHG emissions contributing to global warming. Governor's Executive Order S-3-05 (2005) required a reduction of GHG emissions to 1990 levels by 2020, and a reduction to 80 percent below 1990 levels by 2050. California's 2006 Climate Change Solutions Act (Assembly Bill 32) made the 2020 reductions law and tasked many government entities, including local governments and government agencies, with helping to meet those goals. Governor's Executive Order B-30-15 (2015) requires a reduction of 40 percent below 1990 levels by 2030, which was codified into law with Senate Bill 32 in 2016. Executive Order B-55-18 (2018) requires carbon neutrality for the state by 2045.

The most recent California Climate Inventory published by the California Air Resources Board (CARB) found that in 2016 the state emitted 429.4 million metric tons (MMT) of carbon dioxide equivalent ($CO_{2-eq}$; calculated by CARB using the global warming potential (GWP) factors published in 2007 by the Intergovernmental Panel on Climate Change Fourth Assessment Report over a 100-year time horizon), achieving a 30 percent reduction from 2005 levels and meeting the 2020 goal of a reduction to 1990 levels four years ahead of time [27,28]. The 2016 inventory showed that the transportation, industrial, and electricity generation sectors of the economy were the three largest sources of emissions, with 41%, 23%, and 16 percent, respectively. The large majority of the emissions in the transportation sector is from combustion of gasoline and diesel, and in the electricity sector are from combustion of natural gas in in-state power plants, and importation of energy from combustion of coal in out-of-state plants. In the industrial sector, a large portion of the emissions are from oil and natural gas production and oil refining, of which a portion can be attributed to an asphalt binder used in transportation infrastructure, and there are contributions from the production of cement and steel used in transportation infrastructure.

There is no single change that will achieve the ambitious goals set by the state for itself, instead multiple changes must be made in the state's economy by many actors. Many changes are being proposed by a multitude of sources, with the proposals based to varying degrees on science,

economics, the potential to grow markets or shrink the markets of competitors, regulatory strategies, and attractiveness often based on the simplicity of the idea and the ability to easily communicate it to policy-makers and the general public. Identifying, quantifying, and then selecting among the many possible strategies to achieve GHG reductions is difficult, especially without a standardized approach for comparison.

Over the last five years, state and local policy leaders and transportation system operators have identified the difficulty of prioritizing the tens of strategies and tactics that are being proposed for changes in how they should design and operate systems to reduce GHG emissions to meet the state's goals. One method of doing so is by considering policy packaging, which capitalizes on the synergistic relationship between various policies and maximizes effectiveness through appropriate bundling. Researchers have proposed a framework to group policies according to this synergistic relationship [29]. Others have developed a ranking method that considers more than just synergistic relationships, such as one-way facilitative relationships and contradictory relationships [30]. One shortfall of these methods is that they are inherently qualitative. For example, the ranking framework developed by Taeihagh et al. [30] relies on, among other things, expected public approval, and the perceived extent to which policies are interlinked. Additionally, they only look at the direct, "use stage" impacts of the projects, which is sufficient for some policies but certainly not for all.

In parallel with integrated packages of policies to address environmental issues, it is the experience of the authors that at the state and local levels there is a stream of essentially ad hoc proposed regulations, laws, specifications, and policies being proposed that are intended to reduce environmental impacts that appear to be beneficial. These proposed changes are often unquantified, and have been considered within narrow system boundaries and without consideration of life cycle implications. The authors have worked extensively supporting state and local government in California with "life cycle thinking" data and tools for implementation of life cycle assessment (LCA) for environmental impacts (including social impacts) and life cycle cost analysis (LCCA) for financial impacts. Therefore, a process is proposed that considers the full system and complete life cycles, as this provides the most beneficial and sustainable solutions while minimizing the likelihood of unintended negative consequences. It is recognized that the impacts and implementation of considered strategies can change based on the concurrent policies. However, the scope of this project is not to propose policy decisions, but to inform them. The final prioritization of strategies will differ across jurisdictions depending on their goals, funds, and timeline.

The focus of this study is to examine two strategic options that the California Department of Transportation (Caltrans) could adopt to lower its GHG emissions in operating the California state highway network so it can help meet the state's climate change goals as examples of the use of the approach used to evaluate them. Although many GHG reduction strategies appear to be attractive, simple, and effective, the following limitations are also true for many of them:

- The net GHG reductions that result from implementing the strategies have often not been quantified;
- Few of the cases where GHG reductions have been quantified have used a system-wide perspective for their estimates;
- The time it will take to implement a strategy and begin achieving GHG reductions has not been considered;
- The process and difficulty of making the change have not been estimated, and;
- Most importantly, the costs, or in some cases savings, of implementing both initial and life cycle strategies have rarely been estimated in a way that prioritizes the most cost-effective strategies that would allow maximal emissions reductions with minimal costs.

A life cycle perspective is required for GHG accounting because benefits achieved during one stage of a system's life cycle may be reduced or reversed by unforeseen increases in GHG emissions simultaneously caused in upstream or downstream stages. Similarly, if an incomplete system view is taken, benefits in one part of the system may be reduced or reversed (i.e., more carbon is emitted than

business as usual) in another part of the system that was not considered. In some cases, two or more potential changes in operations are incompatible with each other in ways that will negate the benefits, and a full system view can help identify these conflicts.

The last point in the bullet list above is considered equally important with the calculation of emissions, because state government and the state's overall economy have a finite capacity and political will to pay for change. The approach used in the studies described in this paper is that the greatest and fastest GHG reduction will occur if there is a prioritization in terms of GHG reduction benefit based on the cost of the proposed system change. Prioritization based on the benefit to cost will result in the most efficient use of existing funds to achieve the maximum reduction possible, in other words the most "bang for the buck". The capacity of the public and the state's economy to implement the needed GHG reductions may be exceeded before the goals are reached without a prioritization of this type. It is also considered important to be able to demonstrate to the public that efforts are being made to achieve GHG reduction goals in the most cost-effective ways possible in order to help maintain public support for those goals.

The ability to quantify the full-system and life cycle effects of decisions and changes in systems is advancing and improving using the LCA approach and related analysis processes. LCA is a structured evaluation methodology that quantifies environmental impacts over the full life cycle of a product or system, including impacts that occur throughout the supply chain. LCA provides a comprehensive approach to evaluating the total environmental burden of a product, examining all the inputs and outputs over its life cycle, from raw material production to the end of the product's life [31]. The limitations and problems with LCA are also being identified so that more robust and trustworthy results can be produced. With regard to LCCA, the methodology is already mature and used within Caltrans for support of decision-making regarding infrastructure choices [32].

The timeframe for change is also important because emission reductions that occur sooner will have greater beneficial impact than emission reductions that occur later or are spread out over a longer period of time. This is not accounted for in current GWP calculations. Time-adjusted warming potential (TAWP, [33]) as well as GWP are used to account for the timing of emission reductions in the approach proposed in this paper. Use of TAWP will help identify strategies providing the "fastest bang for the buck".

This study discusses and shows early examples of use of a GHG reduction "supply curve" framework to support decision-making by Caltrans. The supply curve, as used in these studies, provides a method for selecting the most cost-effective strategies for GHG reductions by undertaking the following process for each strategy: (1) quantification of the net effects on GHG emissions over the strategy's lifecycle, (2) consideration of the time required to make the change happen, (3) exploration of the change process and difficulty of making the change happen, and (4) calculation of the initial and lifecycle costs of the strategy. This approach is currently being used for two studies:

1.  To evaluate possible changes that Caltrans can make in its operations to reduce GHG emissions
2.  To evaluate proposed actions for transportation in climate action plans that have been developed by cities and counties in California to reduce GHG emissions.

## 2. The Approach

The approach used was to support strategic prioritization of alternative changes for reducing GHG emissions using what are called "marginal abatement curves", "supply curves", or "McKinsey curves" after the company that has made extensive use of them [34]. Supply curves illustrate the economics associated with changes and policies made for climate change reduction. In particular, the work done by Lutsey and Sperling [35] demonstrated how alternatives within the transportation sector could be quantified and compared using the available information, and also compared with alternatives in other sectors of the economy. Transportation is particularly important in California because it is responsible for approximately 41 percent of annual GHG emissions in the state. This percentage has increased and

actual transportation emissions have increased as other sectors of the economy, particularly generation and use of electrical energy, have decreased [28].

A generic example of a supply curve, recreated from Lutsey [36], is shown in Figure 1. To implement the development of supply curves, a set of questions are answered, and calculations are completed using the best available information about the proposed changes, with each proposed change shown as a box on the plot (the complete set of questions and calculations are described later in the paper). The supply curve uses the best estimate of the benefit on the x-axis, with each box representing a proposed change and the width of the box indicating the size of the benefit. Reduction of GHG emissions is shown in the example, however this could be a performance metric for other environmental goals, such as air pollution, which is a major concern in California, as well.

The y-axis of the supply curve shows the cost of the change per unit of benefit. Two values are calculated for each proposed change using the best available information: the initial cost of implementation and the long-term or life cycle cost (LCC). As with the LCA information, the economic analysis of the proposed changes for the supply curve is developed with the best available information and documentation is required of the assumptions, calculations and quality of the information used.

The proposed changes are put in rank order of cost effectiveness, with color coding to identify the level of uncertainty of the information used for the analysis (not shown in the example in Figure 1). All changes have an implementation cost, indicated by the unpatterned portion of the box, but some changes will potentially result in an LCC savings. Those changes that are to the left on the curve should be considered for implementation first, because they provide the most improvement for the least cost. Those that have negative LCCs are what Lutsey [36] refers to as "no regrets" choices because they reduce costs over the life cycle. Moving to the right along the x-axis of the curve identifies the cumulative effect of changes towards the overall GHG reduction goal, and the increasing cost of achieving that goal. As with all economic analyses regarding public policy, the economic analysis should consider not only the overall costs, but who pays the costs or receives the savings, and whether those costs or savings are equitable.

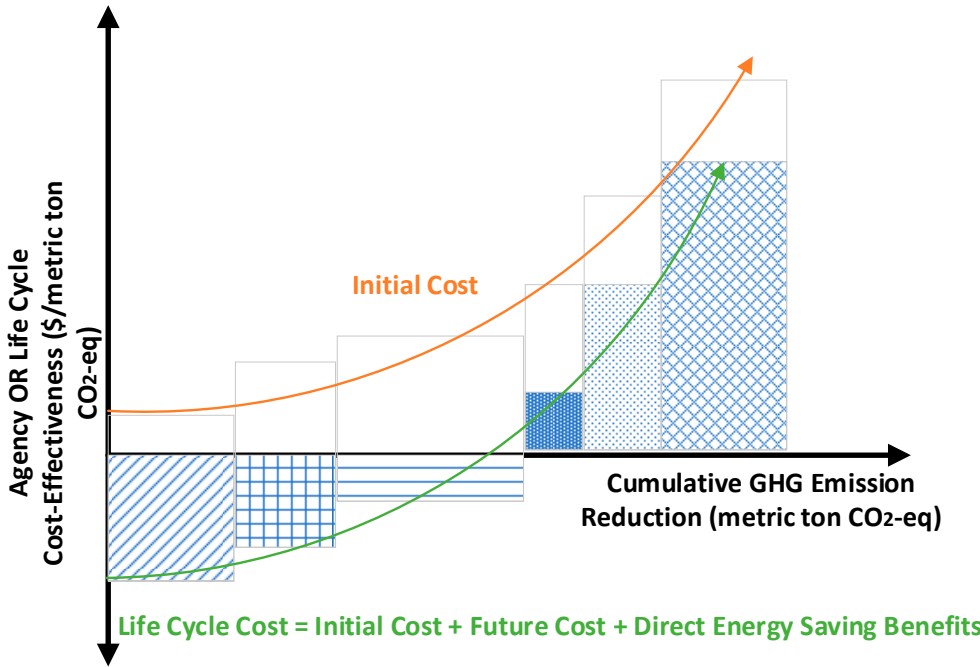

**Figure 1.** Generic supply curve considering initial cost and life cycle cost (recreated Lutsey's curve [36]). (Note: the patterns in the figure differentiate the strategies from each other for better visuals in a grey print and each pattern presents the life cycle cost effectiveness (agency + user) of each strategy).

The purpose of developing supply curves to review alternatives is to bring a full system analysis, life cycle thinking, and above all, quantification, to their development in a decision-making environment where they are often absent, and to support decision-making for prioritization that includes consideration of economics.

## 3. Methodology

In the approach being used, LCA was used to estimate the benefit by comparing GHG emissions from the proposed change over the life cycle analysis period versus current practice. The LCA was performed using the best available information, which could range from very poor to very good based on ISO 14044 [37] data quality parameters as discussed related to pavements in the Federal Highway Administration Pavement LCA Framework [31]: time-related coverage, geographical coverage, technology coverage, precision, completeness, representativeness, consistency, and reproducibility. The documentation of the LCA for the supply curve includes a data quality assessment, which must be taken into consideration when comparing alternative proposed changes on the supply curve.

The details of the process for developing the supply curves included in this paper consisted of:

- Definition of the functional unit and system boundaries for the technology;
- Identification of available information;

  ○ Technology of the strategy;

    ■ Initial implementation;
    ■ Life cycle, including maintenance, rehabilitation, replacement or end-of-life.

  ○ Costs of the strategy;
  ○ Constraints on implementation relevant to implementation by Caltrans.

- Creation of information;

  ○ By analogous estimating from existing sources about similar technologies, different scales of research, development or implementation, or implementation in different contexts;
  ○ By bottom-up estimation from existing sources about components of the technology.

- Calculations;

  ○ Life cycle inventory and impacts;
  ○ Initial costs;
  ○ Life cycle costs.

- Assessment of data quality;
- Inclusion of the strategy on the supply curve.

The best available information that could be obtained was used for both the LCA and the LCCA. Data quality for both was assessed as part of the data analysis. There was a wide range of data quality across the different strategies, primarily dependent on the maturity of the research and development for the strategy, and the amount of implementation that has occurred to provide real world costs.

A questionnaire was developed to help the analyst develop the LCA and LCCA analyses for the supply curve and to also develop information regarding the potential for implementation, including the definition of the technology and the system it would change, the state of readiness of the strategy, the responsible stakeholders, and the factors that would drive the change. The questions to support the LCA and LCCA are standard practice in these methods. The questions used to identify, document, and analyze the implementation of the proposed change come from the experience of the authors

working with state and local government on the implementation of a wide range of technologies, primarily for transportation infrastructure, and from critical evaluation by the authors of climate action plans produced by local governments. They were not based on the literature of large-scale policy development. The questions in the questionnaire were:

1. Define the change/technology; this question requires the proposer to specifically define the proposed change;
2. Define the state of readiness of the change of technology using approach adapted from the NASA Technology Readiness Level system [38]; decision-makers are often not aware of the readiness of the proposed technology;

    a. TRL 1: basic principles observed;
    b. TRL 2: technology concept formulated;
    c. TRL 3 and 4: experimental proof of concept/ technology validated in lab;
    d. TRL 5 and 6: technology validated or demonstrated in relevant environment at less than full scale (industrially relevant environment in the case of key enabling technologies);
    e. TRL 7: system prototype demonstration in the operational environment (full scale);
    f. TRL 8: actual system completed and "flight qualified" through test and demonstration;
    g. TRL 9: actual system proven in operational environment elsewhere or less than full market penetration.

3. Define the system in which the change occurs this information defines the specific context in which the change is proposed to be implemented;
4. Will the market change or is it just changes in market share; this information addresses the approach to be used for the life cycle assessment, either attributional (changes within the existing market) or consequential (new markets appear and/or old markets disappear or are fundamentally changed);
5. Who is responsible for the change; implementation is often stopped because the responses of critical stakeholders, particularly those who must make changes, are not explicitly identified and resistance or buy-in planned for;
6. Who is responsible for implementing the change; most change requires a champion to push it, unless it is completely market driven;
7. Who pays for the change (this information is needed to identify the financial capacity and willingness of those responsible to pay for change):

    a. Government, level of government;
    b. Producers without pass through to consumers;
    c. Consumers.

8. What will drive the change (this identifies the implementation approach):

    a. Market;
    b. Market incentives (example, tax break);
    c. Regulation;
    d. Legislation;
    e. Public programs incentivizing change;
    f. Education;
    g. What will the change do to these other environmental indicators (this identifies unintended consequences in other impact areas, particularly when the proposed change has one specific goal such as GHG reduction):

        i. Air pollution;

ii.　　Water pollution;

iii.　　Energy use:

- Renewable;
- Non-renewable;
- Renewable energy source used as material;
- Non-renewable energy source used as material.

iv.　　Water use;

v.　　Use of other natural resources.

9.　　What are the performance metrics; this information is needed to assess progress and success during implementation, and make required changes in the implementation strategy if needed;

10.　　Supply curve calculation questions (the results needed to build the supply curve):

a.　　Expected change in GHG output per unit of change in system;

b.　　Expected maximum units of change in system;

c.　　Time to reach maximum units of change;

d.　　Expected rate of implementation;

e.　　Total estimated initial cost (to be used with total change in GHG to calculate initial cost per unit of change);

f.　　Estimated LCC per unit of change (to be used with total change in GHG to calculate the initial cost per unit of change).

11.　　Methodology for developing information to answer questions; this information is needed so that critical reviewers and stakeholders can review, understand, and critique the supply curve and implementation plans;

12.　　Any available documentation for answers to all questions; this is documentation needed for the transparency of scope, goals, methodology, data, and data quality;

13.　　Data quality assessment; this information is needed for decision-makers and stakeholders to understand the limitation of the quantitative information used for the supply curve, and can also be used to identify where additional effort should be made to develop better data for promising proposed changes that have high uncertainty;

14.　　Critical review of results; this is documentation of the critical review.

The information used to develop the answers to all questions needs to be fully documented, including:

- Citations;
- Development of optimistic, best, and pessimistic estimates to the extent possible to permit sensitivity analysis; to help assess the robustness of the supply curve information;
- Identification of the level of disagreement between different sources of information; needed to help assess the robustness of the supply curve information;
- A ranking of the data and estimation quality such as excellent, good, fair, poor, and completely unknown; needed to help assess the robustness of the supply curve information.

The recommendation is to submit supply curves and their documentation to outside critical review by interested stakeholders before using them for decision-making and documentation of the critiques and responses by the supply curve developers, following ISO LCA principles.

The information collected in the framework includes both quantitative and qualitative data, which decision-makers are expected to use in a process of multi-criteria alternative prioritization. The supply curve provides the primary data, providing a rank ordering of strategy cost-effectiveness in terms of cost per unit change of impact, and quantification of contribution towards achieving impact

reduction goals. However, to be implemented, strategies must be ready for implementation, and those that are not yet developed to a point at which they can be implemented have more uncertainty in the quantification of their costs and impact changes and will need further investment in development. Information to assess technology readiness comes from the NASA Technology Readiness Level. Decision-makers should consider TRL when selecting from the most promising strategies on the supply curve, both in terms of risk of achieving the expected outcomes, and time to achieve those outcomes.

For a strategy to be implemented, the system in which the implementation will occur, what implementation will change in the system, the process of implementation (who is responsible, the mechanism (regulation, market, etc.), and funding) need to be understood. The framework calls for this information to be included in the decision making so that planning, including risk assessment, of the expected approaches or alternative approaches for implementation can be done. The performance metrics for the evaluation of the success of implementation also need to be identified in the framework. If any of this information cannot be identified, then the strategy is not viable.

As noted at the introduction to this section of the paper, this methodology was based on the experience of the authors supporting local and state government decision-making, and was intended to be used by state and local government to quantify changes in environmental impact from proposed changes, and help them identify, document, and analyze a proposed path for implementation. The example case studies presented in this paper were analyzed at the proposed implementation scale for the state of California. Smaller scale analysis should be done for specific local governments, as the quantitative information needed for the approach is often context-specific. The proposed methodology was not intended to be used for simple scaling up for integrated national or international policy development. However, it is likely that this approach of quantitative evaluation of proposed changes at the scale of implementation could provide useful information for a bottom-up building of integrated policy plans, provided appropriate inter-system effects were considered.

## 4. Applications in Studies Currently Underway

This approach was currently piloted for proposed changes in the operations of the California Department of Transportation (funded by Caltrans), and for alternative strategies being included in climate action plans under development by California local and regional planning agencies (funded by the National Center for Sustainable Transportation (NCST)).

For the Caltrans study, the above methodology was currently being applied as a pilot to six GHG reduction strategies that could be implemented by Caltrans. These strategies were selected to provide a wide range of topics with which to test the evaluation process:

1. Efficient maintenance of pavement roughness;
2. Energy harvesting through piezoelectric technology;
3. Automating bridge tolling systems;
4. Increased use of reclaimed asphalt pavement;
5. Electrification for light vehicles and use of bio-based diesel as alternative fuels for the Caltrans fleet, and;
6. Installation of solar and wind energy technologies within the state highway network right-of-way.

The best available information that could be obtained was used for both the LCA and the LCCA. Data quality for both was assessed as part of the data analysis. There was a wide range of data quality across the different strategies, primarily dependent on the maturity of the research and development for the strategy, and the amount of implementation that has occurred to provide real world costs.

A description of two of these potential changes, strategies 1 and 6 in the list above, and initial findings from the study that was currently underway were described below. The analysis period was for 30 years. The analysis period was based on the executive order mandating carbon neutrality by 2045, which was recently signed by the governor of California. The impact indicators were calculated using

the TRACI impact assessment methodology, the most commonly used selection of impact categories in the U.S. The TRACI methodology was developed by the United States (US) Environmental Protection Agency (EPA), and the most recent version (TRACI v2.0) was released in 2012 [39,40].

### 4.1. *"Optimized" Triggering of Pavement Roughness to Reduce GHG*

Pavement condition affects the fuel use of vehicles and therefore both GHG emissions and the cost of transportation, while maintaining pavement condition is a direct cost to road agencies. Pavement condition affects the fuel use of vehicles through rolling resistance, i.e., energy losses due to interaction between vehicles and the pavement. The relative impact of the three elements of rolling resistance (roughness, texture, and structural response) on fuel economy and GHG emissions from on-road vehicles was assumed in this study to depend primarily on the level of pavement roughness in California considering the current level of roughness on the network. The effects of the structural response were the subject of a current study by the University of California Pavement Research Center (UCPRC) for Caltrans. Vehicle fuel consumption is most efficient on smooth pavement surfaces. Evans et al. [41] showed that a decrease of 10 percent tire rolling resistance could increase fuel economy by approximately 1.1 percent. However, in order to keep the surface smooth, maintenance and rehabilitation (M and R) treatments are required, which demand additional resources and energy, thus producing GHG emissions.

### 4.1.1. Study Scope, System Boundary and Functional Unit

This strategy involves changing the prioritization of M and R treatments in the decision trees of the Caltrans' pavement management system (PMS) to include consideration of the reduction in GHG by use of international roughness index (IRI) trigger levels "optimized" for different levels of traffic, instead of the current prioritization based primarily on the preservation of the pavement infrastructure and avoiding excessively rough pavement.

A single IRI triggering value can be used, as adapted by Caltrans, for M and R over the entire highway network. Caltrans changed their single IRI triggering value for M and R in the PMS from 3.45 m/km (220 in/mile) to 2.68 m/km (170 in/mile) in 2010. It is hypothesized that maintaining the roads in a smoother condition (keeping roughness lower) may reduce life cycle GHG emissions as well as costs. This can be achieved by dividing the road network into segments based on the traffic characteristics and then identifying an "optimized" IRI trigger value per road segment. Optimized is stated in quotation marks because the optimization included in this study was empirically derived from simulations rather than a formal closed-form optimization process, and because there were limitations in the scope of the optimization exercise, described later.

The state highway network managed by Caltrans includes 77,685 lane-km (approximately 47,954 lane-miles) of pavement, managed using the Caltrans PMS. The network is approximately one quarter concrete surfaced pavement and the remainder asphalt surfaced pavement. The concrete pavement consists primarily of jointed plain concrete, much of which was not built with dowels (construction prior to 2000), and some newer continuously reinforced concrete. The asphalt surfaced pavements include flexible (asphalt on granular base or subgrade), composite (asphalt on concrete), semi-rigid (asphalt on cemented base), and some segments that have had full-depth reclamation (FDR) and cold in-place recycling (CIR). Treatments for concrete pavement include slab replacements, grinding, slab replacements with grinding dowel bar retrofit, concrete lane replacement, concrete overlays, and asphalt overlays. Treatments for asphalt surfaced pavements include seal coats, thin to thick asphalt overlays, FDR, and CIR.

GHG emissions were calculated for these treatments in the PMS using materials and construction emissions factors developed by Wang et al. [42,43], which are shown in Table A1 (for asphalt surface treatments) and Table A2 (for concrete surface treatments) in the Appendix A. Table A3 in the Appendix A presents the coefficients that are used in the GHG equation for IRI. The equations used for GHG emissions calculations in PA® from IRI (Equation (A1)) and, materials and construction

(Equation (A2)) are presented in the Appendix A. The timing of M and R treatments is based on cracking and IRI progression models and initial IRI values for treatments developed by the University of California Pavement Research Center (UCPRC) for Caltrans and implemented in the PMS [44]. The implementation of these models in the PMS code has been checked and confirmed by Caltrans and the UCPRC. Decision trees for concrete and asphalt surfaced roads trigger a treatment in the pavement management system analysis of future treatments over the forward time horizon based on predicted development of cracking and roughness from these models.

The current PMS decision trees consider cracking first, and then an IRI of 2.7 m/km (170 inches/mile) to trigger treatment, with the same roughness trigger applied to the entire network. The alternative considered in this study was the use of IRI triggers that focus on keeping sections with higher traffic volumes smoother to maximize the reduction in GHG. The optimized trigger levels developed by Wang et al. [42,45] balance greater material and construction emissions from performing an M and R treatment and the fuel savings per vehicle resulting from smoother pavement multiplied by the number of vehicles using the pavement to minimize the sum of the maintenance and construction plus the use of stage emissions. The GHG emissions from M and R materials and construction need to be paid back through reductions in GHG from the vehicles operating on the smoother pavement to be effect, and there is a traffic volume level at which they will not be paid back. Pavement with very high traffic will have maximum reduction of GHG with pavement kept smoother than will pavement with lower traffic levels. At the same time, higher truck traffic flows accelerate pavement deterioration, which requires more frequent maintenance in order to keep the network functional.

The interactions of these models and the net results had been explored in the previous work, however, this study is the first time that they have been applied to the entire network over a long time horizon, and in the production-level analysis of the PMS. The "fine" segmentation of the network was used, which divides the network up, lane-by-lane, into segments with similar traffic, climate, pavement structure, condition, and past construction history. Segment lengths are mostly less than one 1.6 km (1 mile).

Two scenarios were evaluated:

1.  Unlimited budget_currentIRI—assumed no budget constraint on M and R activities. Current Caltrans decision trees were used, which trigger M and R based on predicted cracking with different treatments for different levels of cracking and faulting, or if the IRI value was 2.68 m/km (170 in/mile) or greater.

2.  Unlimited budget_optimizedIRI—used the same assumptions as scenario 1, except that an optimized IRI trigger value was used to trigger a treatment if not already triggered by cracking:

    *   Segments with passenger car equivalent (PCE) per day less than 2517—no IRI trigger value (no treatment);
    *   2517 < PCE ≤ 11,704—IRI trigger value of 2.8 m/km (177 in/mile);
    *   11,704 < PCE ≤ 33,908—IRI trigger value of 2.0 m/km (127 in/mile);
    *   33,908 < PCE—IRI trigger value of 1.6 m/km (101 in/mile).

PCE converts the trucks to the equivalent number of passenger cars that can displace the truck. The value of 1.5 was used in this study. Approximately 30 percent of the network had PCE per day greater than 33,908, and approximately 25 percent of the network had PCE less than 2517.

The functional unit for this study was defined as the M and R of the California state highway network maintained using the decision trees defined in scenarios 1 and 2, for an analysis period of 30 years. Materials and construction for the M and R stages, and vehicle fuel use as changed by pavement roughness were included in the system boundary.

### 4.1.2. Assumptions and Limitations

The major assumption for both scenarios is that the state funding for M and R determined by the decision trees is not constrained by budget limitations. Historically, this has been far from the reality. However, recently passed Senate Bill 1 increased the state motor vehicle fuel tax, and the law was confirmed by direct public vote in November 2018. The additional funding includes approximately $1.5 billion for the State Highway Operations and Protection Program that are to be used for M and R on the state highway network, and consideration of an unlimited budget helps identify what the funding needs would be to perform all of the treatments called for by the decision trees of the two scenarios. The emissions factors for the treatments are based on data from the literature and new research by the authors and colleagues that were available in 2013. The details of emission factors calculations based on a developed relationship between the roughness equation from HDM-4 model and vehicle emissions equation developed from MOtor Vehicle Emission Simulator (MOVES) can be found in References [45,46]. The sources of those data and an assessment of the data quality are shown in Appendix A. It was assumed that these values are reasonable and acceptable for conceptual-level analysis, although some of the information noted in those references were based on older European sources. Those inventories were currently being updated but would not be updated in the PMS until 2020, after which the analysis would also be updated. Life cycle inventories for pavement were expected to go through extensive updating in North America over the next five years.

Another limitation was that the original scope for the development of the optimized IRI trigger values used in scenario 2 was applicable to 2010 to 2013 when Caltrans did not have sufficient funds to do many rehabilitation projects and a restricted set of treatments most commonly being used at the time was considered in the optimization. These were thin and medium thickness asphalt overlays for asphalt surfaced pavement and slab replacement using rapid strength concrete followed by diamond grinding for concrete pavement, along with a few concrete lane replacements using ordinary Caltrans paving concrete for badly damaged segments. The emissions factors account for the fact that Caltrans uses a significant amount of rubberized asphalt mixes for its asphalt overlays, and supplementary cementitious materials in its ordinary paving concrete, which are considered in the GHG emissions factors.

The effects on fuel economy and therefore GHG emissions of change in vehicle speeds, construction work zone (CWZ) effects on congestion, and the interactions of roughness and traffic driving cycles as opposed to the free-flow conditions under which the fuel economy models were developed are other important factors that were not considered in this study. In the analysis, changes in traffic flow in CWZ during treatment construction that might impact GHG emissions were not considered, and it was assumed that work was performed during the night time so no CWZ traffic delay occurred. While Caltrans performs that majority of construction work using nighttime closures is not always the case. A CWZ can increase or decrease emissions compared to conditions when there is no CWZ on a given segment. When traffic is forced to operate with congested stop–start driving cycles then there are often increased GHG emissions, although again the increase is dependent on the speeds without the CWZ. If the CWZ slows traffic to a steady speed of about 75 km/h (45 mph) from higher speeds then GHG emissions are reduced.

It is also assumed that there were no changes in vehicle speeds due to drivers going faster on smoother pavement after the treatment under free-flow conditions than they were on the rougher treatment before the treatment. The literature up to 2013 regarding driver behavior was primarily based on statistical analysis without considering before- and after-treatment measurements for the same location, and showed that drivers moved at higher speeds on a smoother pavement. On the other hand, the results of analysis on California freeways for before and after treatment at the same locations is documented in a previous UCPRC report [47,48], which showed less sensitivity of driver behavior to pavement smoothness (0.3–1.0 km/h (0.2–0.6 mph)) change for typical changes in IRI) than previous studies. In general, increases in free-flow speed above 75 km/h will increase fuel use and GHG emissions. The effects of speed on fuel use and therefore GHG emissions differ from vehicle

to vehicle, under different air temperatures, which affect the aerodynamic drag effects of speed, and under different congestion conditions. A rough estimation from the literature [49] is that an increase of speed of 1.6 km/h (1 mph) will change $CO_2$ emissions from automobiles by about 2.4 percent at high free-flow speeds, while truck fuel use can be much more sensitive [50]. Trucks and automobiles were not separated in the previous UCPRC traffic speed study, however, the outer truck lanes showed less sensitivity of speed change to change in roughness than the inner automobile lanes. There is a range of values for the effects including as low as 0.3 percent change in automobile fuel use per 1.6 km/h increase in speed [51].

The effects of roughness on fuel use under conditions other than free-flow has not been explored in the literature. The models used in this strategy analysis were developed under free-flow driving conditions, and it was assumed that roughness would have the same effects under different drive cycles.

Although small, these changes in speed were of a similar order of magnitude to changes in fuel economy from changes in IRI and should be included in future updating of optimized IRI trigger values and GHG calculations in the Caltrans PMS. Much of the traffic in California occurs in conditions that are not free-flow, and the effects of change of roughness on change of speed under those conditions should also be explored.

The study was limited to one environmental impact, i.e., GWP. The scope of the study could be broadened to include other environmental impacts such as noise, particulate matter (PM2.5), water, and others. There are several other social and cost indicators that could also be included in such an analysis. In terms of cost, the study used agency cost only for the analysis. User costs could give a better picture of how the total life cycle costs are affected. Other costs such as insurance, vehicle damage, and risk costs were not considered in the analysis. Few issues related to social, environmental as well as cost considerations were identified below that could be considered as recommendations for future expansion of the system boundaries of such studies. Other system considerations not included in the system boundary included safety, vehicle depreciation/damage considerations, job creation, noise, accidents, freight damage, applicability (available funds, practicality), effect on vehicle life (less damage/longer life), cost of risk (causality/loss cost), and effect on market (more vehicles on road).

The study assumed that the vehicle fleet did not change over the 30-year analysis period, and had the same composition of internal combustion engines and alternative energy vehicles as in 2015. CARB's emission factors (EMFAC) model shows that in 2015 there were approximately 27 million registered motor vehicles in California, out of which 98,020 were alternative fuel vehicles (AFV; electric and natural gas fueled vehicles) [52]. Thus less than one percent of the state's light vehicles use fuels other than fossil fuels alone, and nearly none of the heavy vehicles in the US use alternative fuels. There is considerable uncertainty at this time as to how quickly the fleet will change to AFVs considering that the current average vehicle replacement time is almost 12 years [53]. It is expected that rough pavements will also increase energy consumption of AFVs, although the impacts of that additional energy consumption will depend on changes in the composition of sources supplying the California energy grid, which also has uncertainty. At this time, there has been no research anywhere in the world regarding the additional energy consumption for AFVs from rough pavements.

Offsetting potential pathways to replacement of fossil fuel burning vehicles is continued growth in vehicle miles traveled (VMT) on the state highway network, which was also not considered in this study because of the large uncertainty regarding how this will change over the next 30 years. After a period of decline starting in 2008 from the long recession, VMT has increased rapidly in the later years of the succeeding economic recovery, driven by increases in population and increasing shortages of affordable housing in exploding job markets causing workers to drive longer distances [54,55].

The results from this study should be considered as preliminary considering these limitations. The intention was to produce an initial indication of the efficacy of reducing pavement roughness on high traffic routes considering the current fleet and traffic, which would be applicable for the early portion of the analysis period, and that could be updated as better information becomes available regarding the extent and pace of change in changeover of the fleet to AFVs, changes in VMT, changes

in the production of electrical energy for AFVs, and the effects of pavement condition on AFV energy use. Calculation methods, data sources, and the data quality assessment are presented in Appendix A. Appendix A also contains the questionnaire answered by the analyst to develop this Strategy 1 (Table A5). The answers were intended to be reviewed during the critical review of the results.

### 4.1.3. Results from Strategy 1

Caltrans' annual costs to perform M and R activities based on the current decision trees with a single IRI triggering value (scenario 1) and decision trees with optimized IRI triggering values (scenario 2) are graphically presented in Figure 2. The initial costs were high in the first two years to eliminate the backlog of triggered segments due to previous highly constrained budgets, followed by several years in which the previously backlogged segments of the network had just been treated and costs were much lower. Over the remainder of the 30 years, there were valleys and peaks in costs for triggered projects as they developed cracking and became rough. The total agency cost for scenario 1 with the current single IRI triggers was calculated to be $21.99 billion, while the cost for scenario 2 with optimized IRI triggers was $22.28 billion. The cost for scenario 2 was $283 million more than for scenario 1. Caltrans used a 4 percent discount rate in LCCA calculations, and that value was also used in this study [56].

Figures 2 and 3 show the annual GHG emissions caused by rough pavement over the 30 year analysis period for scenario 1 and 2 resulting from M and R activities and the use stage (vehicle operation), respectively. The emissions for both scenarios follow similar patterns as the sets of recently treated pavements reached treatment triggers together, like "baby booms" of human population that age together and require similar care at different stages of life. The peaks were somewhat higher for scenario 2 because the high traffic segments were triggering earlier than they would in scenario 1, which was offset by lower spending for scenario 2 in the years after the peaks as the high traffic segments are reaching the scenario 1 IRI trigger level. All segments were also subject to being triggered by cracking.

Most of the benefit of GHG reduction could be seen in the use stage (Figure 3) as less excess fuel was consumed due to smoother pavements. The total excess GHG emissions produced for scenario 1 were calculated to be approximately 2642 MMT and for scenario 2 to be 2631 MMT over the 30 years analysis period. The net result was a difference of 11.7 MMT less GHG emissions for the optimized IRI decision trees compared with the current decision trees, with 0.2 MMT coming from a very small difference in M and R activity emissions, and 11.5 MMT coming from differences in the use stage. The small difference in M and R emissions was likely due to some use of less GHG intensive treatments at earlier stages of deterioration for scenario 2.

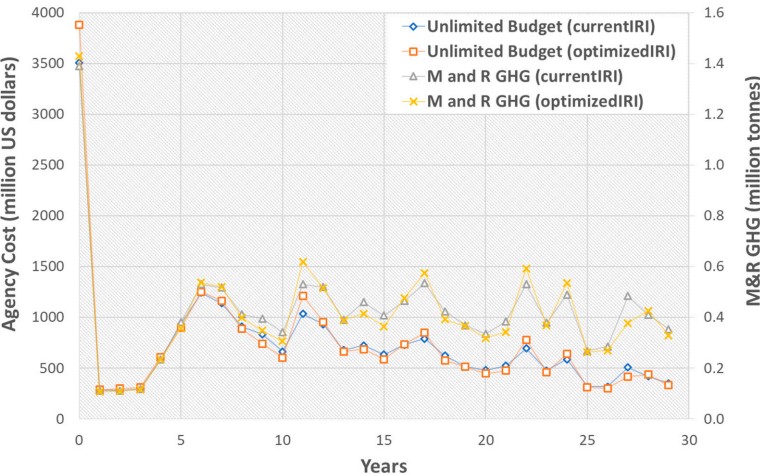

**Figure 2.** Annual agency cost (million $; 4% discount rate) and greenhouse gas (GHG) emissions due to M and R activities versus analysis time period for the two scenarios.

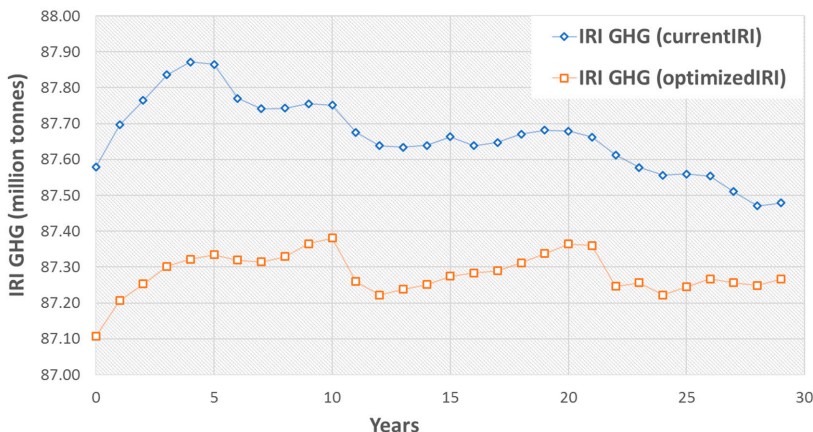

**Figure 3.** Annual GHG emissions during the use stage due to pavement roughness for scenarios 1 and 2.

### 4.1.4. Abatement Potential of the Strategy

When comparing scenario 2 (optimized IRI trigger) to scenario 1 (current IRI trigger), there was a potential to reduce the total GHG emissions (GHG from construction and vehicles) in the range of 0.2–0.55 MMT annually over the 30 year analysis period as shown in Figure 4. The agency cost was calculated to be $24.3 per tonne of GHG emissions if scenario 2 was implemented. This cost was similar to the California–Quebec Cap-and-Trade auction prices since 2012, which have been between $10 and $20 per tonne [57]. However, it is to be noted that the vehicle speed effects on fuel economy and user costs were not considered in this analysis. It was expected that even slightly increased free-flow vehicle speeds would result in some increases in GHG emissions from a smoother pavement. On the other hand, the decrease in user cost due to smoother pavements might be significantly greater than agency costs, which from that wider perspective would make the GHG effects more cost-effective.

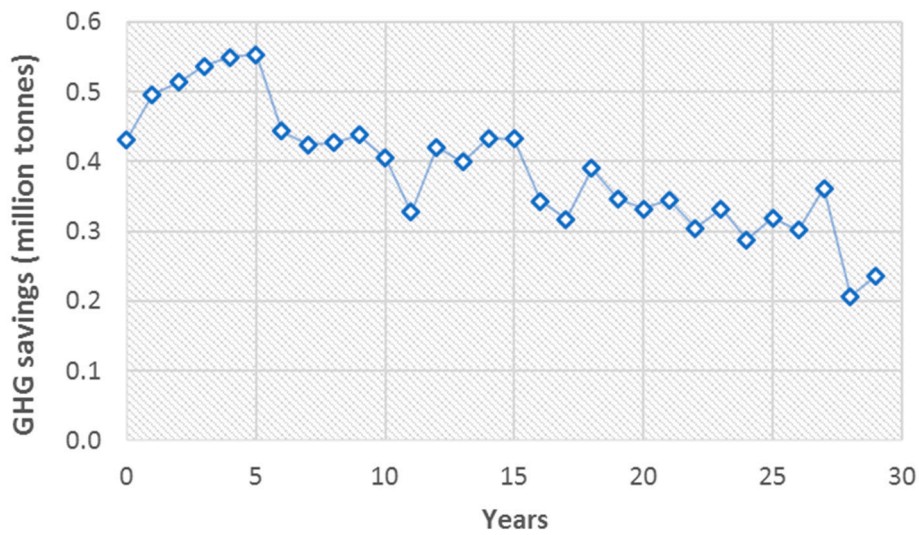

**Figure 4.** Annual GHG savings over 30 years analysis period if scenario 2 is implemented.

### 4.1.5. Time-Adjusted GHG Emissions

Adapting the optimized IRI triggering values (scenario 2) would result in about 2631 MMT of GHG emissions over the 30 years analysis period. However, using TAWP [33] instead of GWP, the resultant GHG emissions for the 30-year time horizon were calculated to be 2330 MMT for the 100 year time horizon. The total GHG emissions reduction due to implementation of scenario 2 versus scenario 1 results was about 10.5 MMT when the time adjusted GHG emissions methodology was used for the 100-year time horizon.

## 4.2. Installing Solar and Wind Energy Technologies within the State Highway Network Right-of-Way

One of the state-wide strategies to reduce GHG emissions from the electrical power sector of the economy is to increase the use of renewable resources, such as solar and wind energy, to reduce the power generation from non-renewable resources, primarily natural gas for power produced in the state and coal for power imported from other states. The state has set electricity generation mix goals of 25 percent renewable energy by 2025 and 50 percent by 2030. The 2016 percentage of renewables on the grid is about 18 percent. Therefore, it is imperative to explore additional possibilities for the installation of renewable energy technologies.

Caltrans owns more than 24,140 km (15,000 miles) of highway centerline, with a large but unknown amount of acreage in the right of ways. Caltrans has already implemented 74 solar projects and has proposed 14 more, but these have thus far been exclusively on buildings according to the Caltrans Sustainability Road Map [58]. No documentation was found online of solar panel installations implemented by Caltrans along the highway right-of-way or as solar canopies, although this idea has been frequently proposed. The first mention of solar panels along the highway right-of-way was in a 2010 presentation exploring this idea on the highways around Sacramento [59]. This idea was mentioned again in a report prepared by a consultant company (ICF International) for Caltrans [60]. In the most recent Sustainability Roadmap, Caltrans frequently mentioned solar canopies as potential GHG reduction strategies [58]. This scenario evaluates the net GHG impacts of generating solar energy on state owned highway facilities (not in the pavement) and wind energy on appropriate locations in the state owned right of ways within freeway interchange cloverleaves.

### 4.2.1. Study Scope, System Boundary and Functional Unit

The proposed abatement strategy is to install small wind turbines in highway cloverleaves (at interchanges), and the installation of solar photovoltaic (solar PV) panels both along state highway right of ways, and in Caltrans owned or operated Park and Ride lots and rest areas. Given the size difference between the technologies, solar PV can be installed along narrower right of ways while wind turbines would require wider right of ways for installation. In this study, the installation of standard crystalline silicon solar cells and 250 kW wind turbines was considered. These wind turbines typically have an average ground-to-blade height of 45 m. These technologies are two of the most mature renewable energy generation technologies that have been implemented around the world.

For wind turbines, junctions and interchanges along the I-5, I-10, and I-15 corridors were considered. In California, these three interstates span 1335 center line miles. For solar panel installations on the highway right of ways, it was assumed that a single row of PV panels were installed in the right of ways along the length of the three aforementioned highways; this number could be smaller or larger depending on (1) where panels cannot be installed, (2) where there is space to install more than a single row at a time, and (3) the spacing required between panels to prevent shading. For solar canopy installations, this study considered Park and Ride parking lots across the California highway network as well as parking lots in rest areas along the I-5, I-10, and I-15 corridors.

The functional unit for this study includes: (1) the implementation of wind turbines at 303 sites in the centers of the clover leaves or in the area available at other types of junctions, (2) crystalline silicon PV cells along the state highway (I-5, I-10, and I-15) right of ways, and (3) PV canopies covering 34,000 parking spaces across Caltrans' Park and Ride lots and rest areas. The analysis period selected was for 30 years. Environmental impacts on GHGs were reported in metric tons (tonnes) of $CO_{2\text{-eq}}$. Materials, M and R, use and end-of-life (EOL) stages were included in the system boundary. Degradation rates (as reported in the calculation methods in Appendix B) of the technologies and an annual discount rate of 4 percent were considered.

The price of generated electricity was uncertain, therefore two prices were used. Under the high price case, utilities provide net-metering benefits, such that the energy generated by the aforementioned installations was used to offset electricity charges to Caltrans by utilities across the state, which includes the electricity used by buildings and to illuminate highways, among others. In this case, a price of

$0.152 per kWh was used, which is the average electricity price across all California sectors according to a report released by the US Energy Information Administration (EIA) [61]. Under the low-price case, utilities purchase the electricity at a significantly lower rate, namely between $0.03 and $0.04 per kWh, as set by the California Public Utilities Commissions [62]. For this case, a value of $0.035 per kWh was used. There are several utilities across the state with each having different electricity price programs, and each can decide to do one or a combination of the above scenarios; therefore, the results provided the range of costs that the strategies would achieve if deployed. A 4 percent annual discount rate was included in cost calculations. Additionally, at the end of the analysis period there were still multiple years left of useful life for all the solar and wind technologies, so their salvage values was accounted for.

4.2.2. Assumptions and Limitations

Tax breaks, rebates, subsidies, or incentives for these installations were not considered. It was also assumed that the installation process of wind turbines and solar panels along the highway right-of-way had insignificant effects on traffic flow and safety. Transmission losses when connecting to the grid were assumed to be negligible. There were several limitations and gaps identified in the study, which included:

- Wind energy potential—a detailed study of wind power potential was not performed, therefore the estimates of power from wind generation would likely be reduced. For this conceptual-level study the national renewable energy laboratory wind prospector mapping tool was consulted, which showed varying potential for wind energy along the three highway corridors [63].
- Additional time required for designing, planning, and permitting—the timelines for the installations of these technologies can vary widely between sites due to differences in landscape, local jurisdiction, available developers, and more. Each site would require its own design and planning, and would then require the appropriate permits. This process can take anywhere between a few months to over a year. However, this study begins the analysis once this process has been completed, and subsequently considers only the installation rate of the technologies.
- Effects of PV glare on driver safety—this is a potential drawback to PV installation along the highway as mentioned in Caltrans' report on strategies to address climate change [60].
- Effects of wind turbine noise on the surrounding community—wind turbines are associated with low-frequency vibrations that have led to complaints from residents who live near them. While it is likely that the wind turbines will be installed in areas with low populations along these largely rural or desert wilderness highway corridors, these effects could also be experienced by drivers, though exposure would be for much shorter periods of time. The specification sheet of the Wind Energy Solutions 250 kW turbine mentions that the noise emissions generated during 8 m/s winds is 45 decibels (dB) at 100 m distance [64]. For reference, the noise level in a library is 40 dB, a quiet rural area's noise is 30 dB (half as loud as 40 dB), and a whisper is 20 dB (half as loud as 30 dB) [65].
- Transmission losses—it is unclear whether transmission losses between the renewable energy generation site and grid are significant; they depend largely on the distance between the installed technology and the nearest grid connection.
- Effects on afternoon ramp load—electricity demand rises sharply in the afternoon and early evening as people return to their homes, and in the summer turn on air conditioning. This coincides with the decreased output of solar energy production. As solar power capacity has increased in California, and particularly from non-utility scale installations, this has led to the requirement for carbon-intensive "peaker" plants, which have often been coal-fired plants in neighboring states, to make up for this difference between supply and demand. Adding more solar energy to the grid could exacerbate this steep ramp-up of carbon-intensive peaker plants, which can result in the unintended consequence of higher carbon-intensity electricity being generated. This reduces the net benefit of supplying solar power, since it must be balanced with carbon intensive peaker

plants. Wind power on the other hand will often ramp up during the afternoons along the targeted highway corridors, although it is also highly variable.

- Urban heat island reduction due to covering building roofs and parking areas—the shading of building roofs and parking areas could reduce the urban heat island effect. This could reduce the amount of energy used for cooling buildings, but could alternatively increase energy use for heating in colder months. Shading of parking lots with solar panels can lower temperatures in parked cars and reduce cooling loads, and potentially increase vehicle heating demand. For vehicles, cooling is a significantly higher energy load than heating, so the net benefit favors vehicle shading.
- Job creation in the renewable energy industry—the installation and maintenance of these technologies would generate jobs, which could be considered as a socio-economic benefit.
- Time of day pricing is not considered—some utilities charge different rates for electricity use depending on which time of the day it is consumed; alternatively, the value of generating electricity during these times is increased, while the value of generation at other times is decreased. For example, the Sacramento-based utility (Sacramento Municipal Utility District), offers time-of-day rates that are higher on summer weekdays from 5 to 8 PM, and lower throughout the rest of the day. This strategy is meant to minimize the afternoon ramp load (as explained above).

The calculation methods, data sources, and data quality assessment are presented in Appendix B. Appendix B also contains the questionnaire answered by the analyst to develop this Strategy 2 (Table A7). The answers were intended to be reviewed during the critical review of the results.

4.2.3. Results from Strategy 2

This study estimated that a single 250 kW turbine had production emissions of 148 tonnes $CO_{2\text{-eq}}$ and had an agency cost of $537,500. Operating at a 25 percent load capacity, this turbine would generate 547.5 MWh in its first year, avoiding 161 tonnes of $CO_{2\text{-eq}}$ based on average electricity generation emissions for California.

An estimated three hundred and three 250kW wind turbines could be installed across highway junctions and cloverleaves, resulting in a total rated capacity of 75.75 MW. One hundred and one turbines are installed every year for the first three years, and are replaced after 20 years. The installation, M and R costs, and salvage value have a net present value (NPV) of $216 million and generate 90,000 tonnes of $CO_{2\text{-eq}}$. Accounting for the emissions reductions benefits achieved by selling the generated electricity to local utilities, this strategy achieves net emissions reductions of 607,000 in tonnes $CO_{2\text{-eq}}$ over the 30-year analysis period. A high electricity price results in a NPV of profits of −$167 million, while a lower price results in a NPV of costs of $119 million.

For solar PV technologies, this study estimated that a 1 kW solar PV system emitted during its manufacture 1.93 tonnes $CO_{2\text{-eq}}$ and costed about $1040; this is in line with the "rule of thumb" cost of $1 per watt. Based on our assumptions, the solar PV system produced 1.64 MWh in its first year, which reduced GHG emissions by 0.39 tonnes $CO_{2\text{-eq}}$. These generation values were used for both highway and canopy installations. The cost for the initial and replacement highway installations was the same as this aforementioned value.

Considering solar PV on the highway right-of-way, 1335 miles of PV panel would provide 307 MW of rated capacity. Full capacity was reached after three years, and the technologies were replaced after 25 years. The installation, M and R costs, and salvage value had a net present value of $361 million and generated 593,000 tonnes $CO_{2\text{-eq}}$. Accounting for the emissions reductions benefits achieved by selling the generated electricity to local utilities, this strategy achieved net emissions reductions of 1,109,000 tonnes of $CO_{2\text{-eq}}$ over the 30-year analysis period. A high electricity price resulted in a NPV of profits of −$900 million, while a lower price resulted in a NPV of costs of $55 million.

For canopy installations, the initial installation cost was $1350 per kW, while the replacement cost was $1040 as mentioned above.

Regarding solar canopy installations, the assumed installation over 34,000 parking spots provided a total rated capacity of 63 MW. Full capacity was reached after four years, and the technologies were replaced after 25 years. The installation, M and R costs, and salvage value would have a net present value of $100 million and generate 177,000 tonnes of $CO_{2\text{-eq}}$. Accounting for the emissions reductions benefits achieved by selling the generated electricity to local utilities, this strategy achieved net emissions reductions of 204,000 tonnes $CO_{2\text{-eq}}$ over the 30-year analysis period. A high electricity price resulted in a NPV of profits of −$157 million, while a lower price resulted in a NPV of costs of $37 million.

Figure 5 shows the net annual GHG emissions for the wind turbines installed on cloverleaves, PV installed along highway's right of ways and PV installed over parking areas over the 30 years time span, while Figures 6 and 7 show the net annual cost value for the installations using high and low electricity prices, respectively. The installations of the wind turbines and solar panels along the highway occurred in the first three years; installation of the solar canopies required four years. Note that the increases in emissions and costs of the turbine installation starting at the 20-year mark were due to replacement of the turbines. For similar reasons, emissions and costs increased for a period after 25 years for the solar installations. The total emissions reductions achieved were proportional to the installed capacity of each strategy.

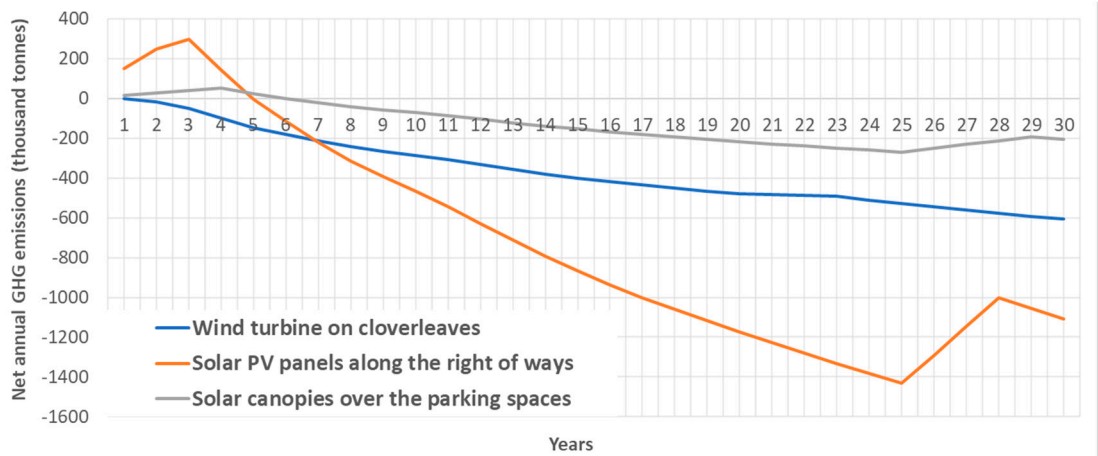

**Figure 5.** Net annual GHG emissions over a 30-year analysis period.

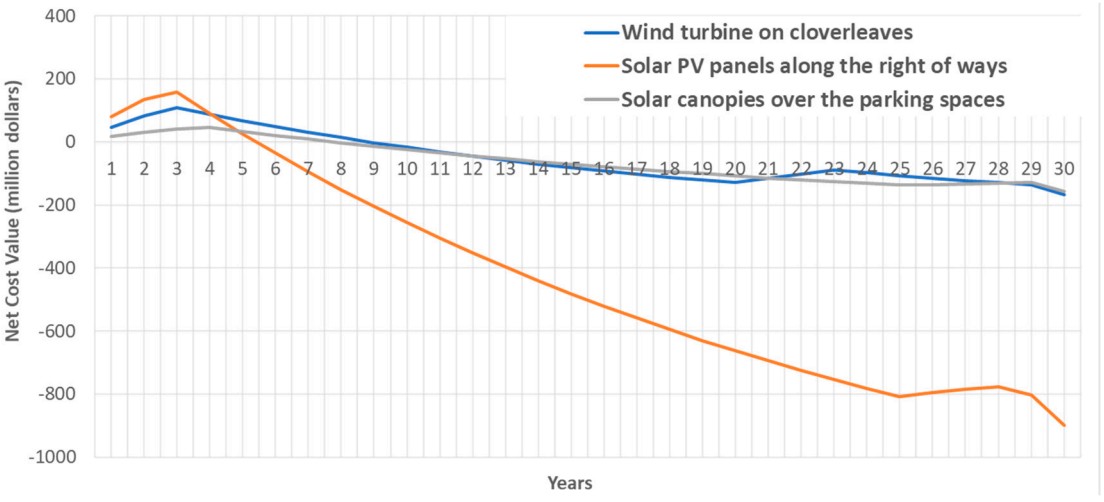

**Figure 6.** Net costs over a 30-year analysis period when generated electricity is worth $0.1535.

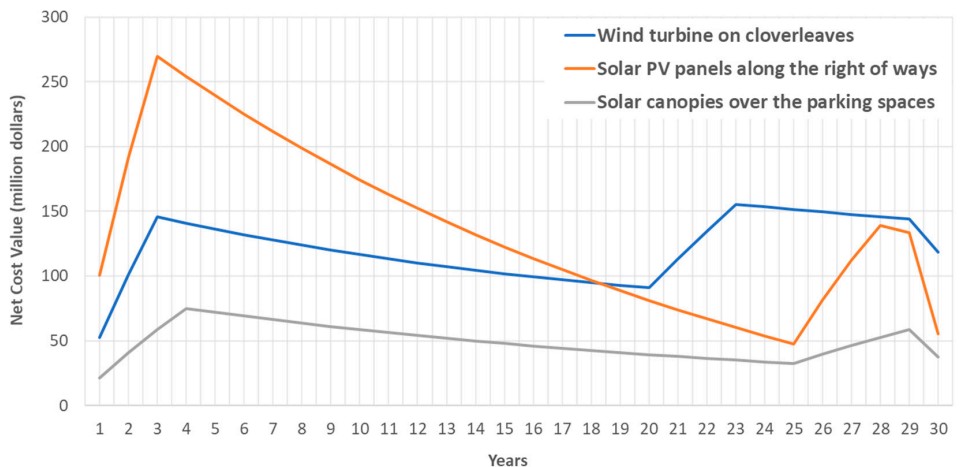

**Figure 7.** Net costs over a 30-year analysis period when generated electricity is worth between $0.035 per kWh.

### 4.2.4. Abatement Potential of the Strategy

The cost effectiveness of the turbine installation considering only the agency cost was $355 per tonne reduction of $CO_{2-eq}$. The LCC effectiveness, which would include income from electricity net metering, was −$275 per tonne reduction of $CO_{2-eq}$ (net savings) in the high price case, and $195 per tonne reduction of $CO_{2-eq}$ in the low price case.

The cost effectiveness of the PV installation along the highway right-of-way considering only agency cost was $325 per tonne reduction of $CO_{2-eq}$e. The life cycle cost effectiveness, which would include income from electricity net metering, was −$811 per tonne reduction of $CO_{2-eq}$ (net savings) in the high price case, and $50 per tonne reduction of $CO_{2-eq}$ in the low price case.

The cost effectiveness of solar canopy installation in rest areas considering only the agency cost $490 per tonne reduction of $CO_{2-eq}$. The life cycle cost effectiveness, which would include income from electricity net metering, was −$771 per tonne reduction of $CO_{2-eq}$ (net savings) in the high price case, and $183 per tonne reduction of $CO_{2-eq}$ in the low price case.

Considering all three strategies, the agency costs of installation and M and R had a NPV of $676 million. The cost effectiveness considering only agency cost was $352 per tonne reduction of $CO_{2-eq}$. A high electricity rates case resulted in a NPV of −$1,211 million (net profits), while a lower price electricity rates case resulted in a NPV of $223 million. The life cycle cost effectiveness, which would include income from electricity net metering, was −$631 per tonne reduction of $CO_{2-eq}$ (net savings) in the high price case, and $116 per tonne reduction of $CO_{2-eq}$e in the low price case.

### 4.2.5. Time-Adjusted GHG emissions

The initial analysis of the turbine installation estimated the net reduction in GHG emissions to be 607,000 tonnes $CO_{2-eq}$. The TAWP [33] 100-year net reduction in emissions was calculated to be 544,000 tonnes $CO_{2-eq}$. For the highway solar PV installations, the initial analysis found the net GHG reduction to be about 1,109,000 tonnes $CO_{2-eq}$, which was 1,009,000 tonnes $CO_{2-eq}$e based on TAWP methodology. For the solar canopy installations, the initial analysis found net GHG reduction to be about 203,000 tonnes $CO_{2-eq}$e, whereas the TAWP was closer to 185,000 tonnes $CO_{2-eq}$.

Proceeding with all three installations reduced GHG emissions 1,920,000 tonnes $CO_{2-eq}$e, but the TAWP was closer to 1,739,000 tonnes $CO_{2-eq}$.

## 5. Supply Curve Information

The two strategies evaluated in this paper could be used to produce a supply curve as shown in Figure 1. For Strategy 1, the total GHG emission reduction was calculated to be 11.7 MMT $CO_{2-eq}$ and the total LCC difference was $24.3 per tonne of reduction of GHG emissions. Accounting for the

emissions reduction benefits achieved by selling the generated electricity to local utilities, Strategy 2 achieved a net GHG reduction of 1.9 MMT. The overall agency LCC effectiveness ranged between −$631 in the high price case and $116 per tonne reduction of $CO_{2\text{-eq}}$ in the low price case.

In terms of prioritization, the results indicate that over the next 30 years, both strategies produced GHG emission reductions. The agency cost for Strategy 1 was low because the optimized Caltrans IRI trigger values did not significantly change the M and R costs compared with the current IRI trigger values. However, keeping the California road network smooth (low IRI based on traffic levels) potentially results in much lower emissions mainly due to lower fuel use by the traffic. The relative costs of Strategy 1 and Strategy 2 were highly dependent on the price Caltrans could get for the electricity it produces, ranging from large cost savings to high costs per tonne reduced, and Strategy 2's cost-effectiveness relative to that of Strategy 1, or any other strategies depended on that price.

## 6. Summary and Critiques of Supply Curves

Governments and road agencies have goals for reducing GHG emissions and other environmental impacts and also face cost constraints. In democracies, there is a need to maintain public support for policies and practices to achieve these critical environmental goals by choosing the most cost-effective alternatives, and honestly, transparently, and effectively communicate the approach used in decision making, the expected benefits and costs, and the metrics for measuring the performance of the decision makers in delivering the results. Many potential changes in the policies and practices of road agencies are being proposed, both internally and externally. However, there is often a lack of quantitative information regarding the benefits and costs of these proposals, and a lack of definition regarding how the changes will interact in a larger system in which they will occur and their long-term effects, and who they will affect, which has equity implications.

A promising approach, called supply curves, that has been used at a national level for developing abatement strategies for GHG reduction with mixed results was proposed for use in this paper to compare strategies at the state and local levels. Some of the critiques of the past use of supply curves were being addressed through the use of the principles of consequential LCA and LCCA. Pilot studies currently underway for a large state road agency and local governments will provide initial feedback on the ability to use this approach at a conceptual level for initial prioritization of alternatives. Initial results indicate that sufficient data could be gathered in a reasonable amount of time to compare alternatives and that the results could be compared on a much more consistent basis than had occurred previously. It is apparent from work to date that a number of important assumptions need to be made, that need to be fully documented, and assessed for quality, for consideration in decision making.

Regardless of whether the reduction in GHG emissions is small or large, every government and industrial sector should find ways to reduce GHG emissions and at the lowest cost possible because cumulative contributions can bring a major change towards a cleaner environment. This paper gave a detailed methodology for developing supply curves for different GHG reduction strategies and presented two examples of strategy assessment for inclusion in a supply curve. A questionnaire was developed that could help in understanding the strategies and became the basis for the supply curve development. The results indicate that keeping the California road network smoother could reduce much more GHG emissions than installing PV panels and wind turbines on Caltrans right of ways. The results also indicate that there was a potential for a net reduction in Caltrans' life cycle operating cost, dependent on the value of the electricity generated.

However, supply curves must be used with caution, and are only one of the tools available to support decision-making regarding GHG and other pollutant reduction, not the only one. A number of limitations of supply curves were identified, including omission of ancillary benefits of GHG emission abatement, poor consideration of uncertainty in the data, a lack of consideration of dynamic interactions over time, and a lack of transparency concerning their assumptions. Supply curves based on the individual assessment of abatement measures suffer from additional shortcomings such as not

considering interactions, non-economic costs, and behavioral changes, as well as incorrect counting of benefits, and inconsistent baselines [66]. It has been suggested that supply curves be used more for comparisons of alternatives than for quantifying cumulative progress to abatement [67]. The ability of supply curves to predict future abatement has been critiqued because of the lack of considerations of longer-term changes in markets driven by consumer changes, the timing of policy actions, actions taken by other actors in the market, and changes in future technologies [68]. Most of these critiques have focused on national-level supply curves, rather than more granular and often less complex curves for agency- and local-level curves, but they must be kept in mind when using supply curves to support decision-making.

These critiques were intended to be addressed somewhat by the use of LCA and LCCA approaches analyzed at the state and local levels, by the additional information that was intended to be gathered as part of the development of the supply curves, and in particular by the use of consequential LCA, which assumed that decisions would result in changes in the market rather than attributional LCA, which assumed that the market would not change.

**Author Contributions:** Conceptualization, J.T.H.; methodology, J.T.H., A.K., A.A.B., M.T.L., A.S., C.K.; software, J.D.L., I.B.; validation, J.T.H., A.K.; formal analysis, A.A.B., M.T.L., J.T.H., J.D.L., I.B.; writing—original draft preparation, J.T.H., A.A.B., M.T.L.; writing—review and editing, J.T.H., A.K.; project administration, J.T.H.; funding acquisition, J.T.H., A.K., A.A.B.

**Funding:** This work was undertaken with funding from the Caltrans, which is greatly appreciated. This document is disseminated in the interest of information exchange. The contents of this report reflect the views of the authors who are responsible for the facts and accuracy of the data presented herein. The contents do not necessarily reflect the official views or policies of the State of California or the Federal Highway Administration. This publication does not constitute a standard, specification, or regulation. This report does not constitute an endorsement by the Department of any product described herein.

**Acknowledgments:** The authors would like to thank Julia Biggart, Tracie Frost, and Rebecca Parker from the California Department of Transportation (Caltrans) Division of Planning, and Nick Burmas and Joe Holland from the Caltrans Division of Research, Innovation, and System Information for support of this project.

**Conflicts of Interest:** The authors declare no conflict of interest. The funders had no role in the design of the study; in the collection, analyses, or interpretation of data; in the writing of the manuscript, or in the decision to publish the results.

## Appendix A

**Calculation methods, data sources, and data quality assessment**

The Caltrans' pavement management system, PaveM, was used to run the scenarios. The data that is embedded in the Caltrans database includes estimated unit agency cost and $CO_2$ emission factors of each treatment including materials, and M and R stages. Some major data sources for pavement include the pavement LCI produced by Stripple et al. [69] in Sweden, and the asphalt inventory produced by the Athena Institute in Canada, EcoInvent (Swiss Centre for Life Cycle Inventories 2011, [70]), the U.S. Life Cycle Inventory produced by the National Renewable Energy Laboratory (National Renewable Energy Laboratory 2011, [71]), and the cement LCI study by the Portland Cement Association (PCA, [72]). Caltrans 2015 accelerated pavement condition survey (APCS) data were used to identify the condition of the road network. In order to have high confidence in the results, a data quality check was necessary. Table A4 shows quality assessment of the data that was used for the analysis. The scoring in the table was based on the recommendations of FHWA pavement LCA framework document and ISO standards [31,37].

**GHG equations and coefficients used in PaveM [73]**

For a given section of pavement the GHG emission amount was determined with the following equation:

$$GHG_{vehicles}(IRI) = \sum_l \sum_k \sum_j \sum_i (f_i \times IRI_k + C_i) \times V_{ij} \times LM_j)_l, \tag{A1}$$



where $GHG_{vehicles}(IRI)$ is the total GHG in tonne of $CO_{2\text{-}eq}$ attributed to IRI, $f_i$ is the roughness factor (also called the IRI factor) representing an average coefficient for converting the IRI to GHG emission for vehicle $i$ due to rolling resistance (expressed in units of tonne $CO_{2\text{-}eq}$/IRI × mile), $i$ = vehicle group (car, two-axle trucks, three-axle trucks, four-axle trucks, and five-axle trucks), $j$ = lane number, $C_i$ = constant for vehicle group $i$ and represents the GHG emission due to other types of resistance that the vehicle's engine needs to resist (in units of tonne $CO_{2\text{-}eq}$/mile), $V_{ij}$ = volume of vehicle group $i$ in lane $j$, $LMj$ = length of lane $j$ in the section, $IRI_k$ = the predicted IRI value in the $k$th year within the analysis period (in units of m/km), and subscript $l$ is the summation index for section number within the analyzed project. The $GHG_{vehicles}$ quantity was calculated in units of tonne of $CO_{2\text{-}eq}$. As can be seen from Equation (A1), for a given segment, the rougher the pavement, the higher the IRI and the greater the amount of $CO_{2\text{-}eq}$ released to the environment due to higher vehicle rolling resistance.

The general form for materials and construction-related GHG emission was determined from:

$$GHG_{materials\&construction} = \sum_l \sum_j (g_j \times th_j \times 12 \times 5280 \times LM_j)_l, \tag{A2}$$

where $g_j$ is the estimated number of tonnes of $CO_{2\text{-}eq}$ per ft$^3$ of the pavement added or removed in lane $j$ in section $l$, $th_j$ is the pavement thickness added (e.g., overlay) or removed (e.g., milling) in lane $j$ in segment $l$ in units of ft, $LM_j$ is the total number of lane miles in lane $j$ in section $l$, and $GHG_{materials\&construction}$ is the total GHG in units of tonnes of $CO_{2\text{-}eq}$ released due to material production and construction for all the given lane miles repaired within the analysis period. In Equation (A2), the lane width was assumed to be 3.66 m (12 ft).

The coefficients used in the equation for a specific treatment type for asphalt pavements are in Table A1, whereas for concrete pavement in Table A2. Table A3 presents the coefficients used for the GHG calculations for IRI.

**Table A1.** GHG coefficient for asphalt surface treatments proposed in the asphalt decision tree (for 1 mile section).

| Treatment Name | Treatment No. | Assumed Treatment Thickness, $th$ (ft) | GHG Coefficient, ($CO_{2\text{-}eq}$ per ft$^3$) | Pavement Analyzer (PA®'s) $GHG_{materials\&construction}$ Coefficient [1,2] |
|---|---|---|---|---|
| "Do-Nothing" | 0 | 0 | 0 | 0 |
| Fog Seal, Slurry Seal, Chip Seal, Seal Coat-Corrective, microsurfacing, Seal Coat-Preventive | 209, 210, 211, 194, 212, 275 | 0.05 | 0.006673 [a] | 21.14 |
| HMA Thin Overlay (th ≤ 0.10 ft), HMA Thin Overlay-Preventive | 195, 276 | 0.1 | 0.006673 [a] | 42.28 |
| HMA Medium Overlay (0.10 < th < 0.25 ft) | 196 | 0.2 | 0.006673 [a] | 84.56 |
| HMA Thick Overlay (th$^3$ 0.25 ft) | 197 | 0.4 | 0.006673 [a] | 169.12 |
| Full Depth Reclamation (FDR) | 199 | 0.57 | 0.006673 [a,b] | 241 |
| Cold In-place Recycling (CIR), Cold In-Place Recycling-Class 3 | 200, 277 | 0.37 | 0.006673 [a,b] | 156.44 |
| Seal Cracks (assumed to have no GHG) | 201 | 0 | 0.006673 | 0 |
| Hot In-Place Recycling (HIPR) | 223 | 0.37 | 0.006673 [a,b] | 156.44 |
| Mill and Fill | 285 | 0.1 | 0.006673 [c] | 42.28 |

[1] see roadway/setup/network analysis/treatments; [2] this coefficient is the product of the GHG coefficient per unit volume, the thickness, the lane width (12 ft), and the unit conversion of mile to ft (5280); [a] back-calculated from a value given in PaveM, which includes $th \times 12 \times 5280$; and [b] FDR assumed to exert same GHG as conventional overlay. The energy of pulverization, mixing, and placement seems to have been ignored or assumed similar to conventional mix. Cold in-place recycling (CIR) also assumed to exert the same GHG burden per volume of material as the conventional hot mix asphalt. Hot in-place recycling (HIPR) assumed to exert the same as the conventional mix per unit volume and assumed to be placed at the same thickness as CIR; [c] fill assumed at 0.10 ft but no consideration for milling depth, removal, transportation of millings, etc. No effect of mill/fill will be observed because of the absence of mill/fill performance model.

**Table A2.** GHG coefficients for concrete surfaced pavement used in the joint plain concrete pavement decision tree (for 1-mile section).

| Treatment Name | Treatment No. | Assumptions | Assumed Treatment Thickness, *th* | GHG coefficient, *g* ($CO_{2\text{-eq}}$ per ft³) | GHG$_{\text{materials\&construction}}$ coefficient, *g* ($CO_{2\text{-eq}}$ per ft³) |
|---|---|---|---|---|---|
| Do Nothing | 0 | 0 | | | |
| Crack Seat and Overlay (CSOL) | 202 | Place one HMA overlay thickness over cracked and seated JPCP slabs | 0.49 ft HMA | 0.006673 [a] | 207.17 |
| PCC Lane Replacement | 203 | Assume slabs (rapid strength concrete) and base replaced. Assume slab thickness and base thickness to be 0.75 ft each. Assume the newly placed base to contribute half of the GHG contributed by the slab. | 0.75 ft PCC, 0.75 ft Base | 0.013760 [b] | 980.81 |
| Grind PCC for Smoothness-CAPM, Grinding-Preventive, Grinding (poor ride only)-Corrective | 204, 283, 284 | Assume grinding depth to be equal to 0.375 inch (0.03125 ft) | 0.03125 ft | 0.00329 | 6.514 |
| Grind/Replace Slabs-CAPM [b] | 205 | Two treatments. Slab thickness 0.75 ft. Grinding depth 0.03125 ft | 0.75 ft PCC slab, and 0.03125 ft grinding | 0.01589 for slab replacement, 0.00329 for grinding | 755.09 for slab replacement to be multiplied by slab percentage needing replacement. For grinding the coefficient is 6.154. (GHG Calculation in groovy script "GHG Calculation") |
| Slab Replacement-Corrective [c] | 206 | Two treatments. Slab thickness 0.75 ft. Grinding depth 0.03125 ft. | 0.75 ft and 0.03125 ft | 0.01589 | 755.09 to be multiplied by percentage of slabs replaced in the lane mile (GHG Calculation and GHG factors in groovy script "GHG Calculation") |
| Groove PCC pavement | 222 | | 0.03125 ft (assumed like grinding) | 0.00329 | 6.154 |
| PCC Overlay | 226 | | 1.12 ft back-calculated from 980.81 and 0.013760 for concrete slab GHG factor. | | 980.81 |
| CRCP Lane Replacement | 247 | Similar to JPCP lane replacement | 0.75 ft PCC, 0.75 ft Base | 0.013760 [b] | 980.81 |
| Dowel Bar Retrofit | 249 | No information available at the moment. | | | |

[a] Crack seat and overlay (CSOL) involves cracking and seating slabs and then placement of a thin leveling course 0.1 ft HMA, followed by a geotextile then HMA overlay from 0.35 to 0.70 ft depending on TI and design life (see Chapter 620 HDM); with a total HMA including the leveling course and a friction course ranging from 0.45 to 0.90 ft HMA; [b] multiply the equation by percentage of third stage cracking in lane that will be replaced; [c] multiply the equation by percentage of third stage cracking for both the grinding and slab replacement contributions.

**Table A3.** The international roughness index (IRI) factor $f_i$ and constant $C_i$ used in Pavement Analyzer (PA®) for the calculation of GHG quantity due to vehicle rolling resistance.

| Vehicle Classification | Roughness Factor (f) | Constant (C) |
| --- | --- | --- |
| Car, Pickup Truck | 0.0098 | 0.36562 |
| Two-Axle Truck | 0.00994 | 1.09834 |
| Three-Axle Truck | 0.02 | 1.80147 |
| Four-Axle Truck | 0.03317 | 2.62255 |
| Five-Axle Truck | 0.03509 | 2.86596 |

1. The new factors are based on T. Wang's factors [43].

2. The factors are used to calculate the $CO_2$ quantity in tonnes per "1000 miles" driven by ONE vehicle of the classes given below.

3. Factors weight-averaged for asphalt and concrete surfaced pavements using 74% versus 26%

4. The original equation for $CO_2$ calculation assumed effect of IRI and mean profile depth (MPD). The MPD was removed from the equations because it was found to have a small effect compared to roughness. The GHG emission was assumed to be solely affected by the rolling resistance associated with IR.

5. The final equation for $CO_2$ quantity is: $[CO_2] = f*IRI + C$, where IRI in m/km and $[CO_2]$ in tonnes.

6. Example calculation: one passenger car driving over a pavement with IRI of 1 m/km (63 in/mile) will produce $0.00980 \times 1 + 0.36562 = 0.37541$ tonnes per 1000 miles driven (i.e., 1000 VMT).

7. If VMT is calculated from directional ADT, then each vehicle class will have its own ADT. The ADT must also be split per lane and then per segment being analyzed. Using segment length $L_i$, ADT on that segment is $ADT_i$, and then $VMT_i$ for that segment is $ADT_i \times L_i$. This will be done for each vehicle class. Then factors f and C for each class are used with each vehicle class. The $CO_2$ is calculated per each one vehicle of each type and then multiplied by $VMT_i$ of each vehicle class and summed over.

**Table A4.** Data quality assessment for Strategy 1.

| Categories | Data Sources | Data Quality | | | | | | | |
|---|---|---|---|---|---|---|---|---|---|
| | | Reliability | Geography | Time | Technology | Completeness | Reproducibility | Representativeness | Uncertainty |
| **Data Type** | | | | | | | | | |
| Lane-miles of state network | Caltrans/PaveM | Very Good | US | Good | Very Good | Very Good | Yes | Yes | Low |
| Pavement types | Caltrans/PaveM | Very Good | US | Good | Very Good | Very Good | Yes | Yes | Low |
| Average pavement thicknesses | Caltrans/PaveM | Very Good | US | Good | Very Good | Very Good | Yes | Yes | Low |
| Annual traffic | Caltrans/PaveM | Very Good | US | Good | Very Good | Very Good | Yes | Yes | Low |
| % vehicle types/class | Caltrans/PaveM | Very Good | US | Good | Very Good | Very Good | Yes | Yes | Low |
| Pavement performance equations (IRI, cracking) | Lea et al. [44] implemented in PaveM | Good | US | Very Good | Very Good | Very Good | Yes | Yes | High |
| Pavement condition (IRI, cracking) | Caltrans APCS data | Very Good | US | Good | Very Good | Very Good | Yes | Yes | Low |
| **LCA Related** | | | | | | | | | |
| Asphalt | Athena Institute [71] | Good | CDN/US | Poor | Very Good | Poor | Yes | Yes | High |
| Cement | Marceau [72] | Good | US | Poor | Very Good | Poor | Yes | Yes | High |
| Other materials | Wang 2013/Stripple [45,69] | Good | SE/US | Poor | Very Good | Fair | Yes | Yes | High |
| Other materials | EcoInvent [70] | Good | SW | Poor | Very Good | Fair | Yes | Yes | High |
| Other materials | USLCI [71] | Good | US | Poor | Very Good | Fair | Yes | Yes | High |
| Materials and treatments factors | PaveM | Good | US | Fair | Very Good | Fair | Yes | Yes | Low |
| **Cost Related** | | | | | | | | | |
| Treatment agency costs | PaveM | Very Good | US | Good | Very Good | Good | Yes | Yes | Low |

**Table A5.** Questionnaire for Strategy 1.

| Question Number | Question | Answer |
|---|---|---|
| 1. | Define change | a. Existing: *current pavement management system decision trees.*<br>b. Change: *Use the optimal M and R timing to minimize the GHG emissions. This would occur in the pavement management system.* |
| 2. | Define the state of readiness of the change of technology (using approach adapted from NASA) | - *TRL 7: system prototype demonstration in operational environment (full scale)* |
| 3. | Define system in which change occurs | - *Caltrans owned and operated state highway network.*<br>- *Manage through PaveM.*<br>- *Cost to be carried within existing budgets unless other funds found.*<br>- *Budget constraint optimization and unconstrained optimization.*<br>- *Cannot be the only criteria for funding M and R.*<br>- *Mostly applicable to high traffic routes.* |
| 4. | Will the market change or is it just changes in market share? | *Not applicable.* |
| 5. | Who is responsible for change? | *Caltrans* |
| 6. | Who is responsible for implementing change? | *Caltrans* |
| 7. | Who pays for change | *State government* |
| 8. | What will drive change (X) | a. Market<br>b. Market incentives<br>c. Regulation X<br>d. Legislation X<br>e. Internal Policy X<br>f. Public programs incentivizing change<br>g. Education |
| 9. | What will the change do to these other environmental indicators | LCA will answer<br>i. Air pollution<br>ii. Water pollution<br>iii. Energy use<br><br>  1. Renewable<br>  2. Non-renewable<br>  3. Renewable energy source used as material<br>  4. Non-renewable energy source used as materialiv.<br>iv. Water usev.<br>v. Use of other natural resources |
| 10. | What are the performance metrics in addition to GHG reduction and cost? | - *Safety changes*<br>- *Measurement of IRI, change of IRI on high vol routes, traffic vols, CWZ, material purchases, travel speed*<br>- *Road user cost* |
| 11. | Supply curve calculation questions: | a. Expected change in GHG output per unit of change in system (LCA).<br>b. Expected maximum units of change in system (LCA).<br>c. Time to reach maximum units of change (reasonable time to be implementable), policy question.<br>d. Expected shape of change rate (dependent on the funding):<br><br>  i. Linear<br>  ii. Increasing to maximum<br>  iii. Decreasing to maximum *(if prioritized)*<br>  iv. S-shaped<br><br>e. Estimated initial cost per unit of change<br>f. Estimated life cycle cost per unit of change (LCCA) |

## Appendix B

**Calculation methods, data sources, and data quality assessment**

Smoucha et al. [74] determined the total emissions produced due to the full life cycle, including manufacturing, transportation, installation, operation, and decommissioning of a wind turbine. The authors presented the emission values for the turbines with rated capacities from 50 kW to 3.4 MW. The emission numbers were checked against the emission values reported by Vestas for a 2 MW Vestas turbine and it was found that the Vestas estimated higher values [75]. Smoucha et al. [74] emission values were also assessed against emissions from a 1.6 MW turbine, and the two were found to be comparable despite there being a size difference. Based on these findings, emissions reported by Smoucha et al. [74] were considered in this study; the lifecycle GHG emissions value of 148 tonnes $CO_{2\text{-eq}}$ for a 250-kW turbine. A capacity factor of 0.25 was considered, which is within the range of the capacity factors used in the other studies that were reviewed. Additionally, EIA found the median wind plant capacity factor in California to be about 0.26 when considering large scale facilities [76]; turbines located near highways are not expected to experience higher capacity factors then ideally located facilities. A study by Staffell and Green [77] found the average performance degradation rate of wind turbines to be 1.6 percent per year, which was accounted for in the performance analysis. Finally, a lifetime of 20 years is used for wind turbines as used in previous LCAs [78,79].

To determine the number of potential sites where wind turbines could be installed, 407 junctions and clover leaves were manually assessed to determine the approximate area available at each site. A study by NREL found the permanent direct land use of wind turbines (which includes the wind turbine, turbine pad, electric infrastructure, access roads, etc.) to 0.75 ± 0.75 acres per MW; in other words, up to 1.5 acres per MW [80]. This upper bound was used in this study, such that each 250 kW turbine was assumed to require 0.375 acres. This number was used to filter out reviewed sites that were too small. Ultimately, there were 303 sites that could potentially accommodate the installation of a 250 kW turbine. The typical installation time of a wind turbine is about 2 months; therefore, it is assumed that across the state, about 101 turbines can be installed per year such that the maximum capacity is reached after three years. Under this assumption, there are 17 "teams" of installers who can each install six turbines per year. This number is used in future assessments of the technology deployment rate.

It was also found from the literature that the capital cost estimates for wind turbines and their installation range from $1 to $2.2 million per MW of rated capacity; this accounts for 80–90% of LCCs, where the remaining cost is for maintenance and repair, and disposal [81]. A separate study on the lifecycle costs of large scale wind projects found the cost of California-based projects in 2016–2017 to be about $2.15 million per MW [82]; these projects exhibit economies of scale that the proposed installations by Caltrans would not, which suggests that the true costs per MW may be higher. This study used a cost of $537,500 per 250 kW turbine.

Most solar PV literature present results of lifecycle GHG impacts in gram of $CO_{2\text{-eq}}$ per kWh, but every study is different due to different assumptions on technology efficiency, irradiance, lifetime, and other factors. A study by Hsu et al. [83] harmonized the GHG values from several studies and found the lifecycle GHG emissions per unit energy to be 52 g $CO_{2\text{-eq}}$ per kWh, including manufacture, transportation, installation, operation, maintenance, plant decommissioning, recycling of materials, and waste disposal. This output was combined with the harmonization assumptions made in the study to find an emissions value of 276 kg $CO_{2\text{-eq}}$ per meter-squared of PV panel.

It is assumed that 100 W solar PV panels measure 39.7 inches by 26.7 inches (surface area of about 0.7 meters-squared) and are arranged vertically (i.e., each panel takes up 26.7 inches parallel to the ground and extends 39.7 inches vertically) to maximize installation density. Installing 100 miles of panel in this orientation provides a rated capacity of 23.6 MW. The power-to-area value above was used to find that the amount of PV panel required for 1 kW of power generates 1.93 tonnes $CO_{2\text{-eq}}$ over its lifecycle; in other words, this is the amount of lifecycle emissions generated by 7 meters-squared

of PV. It is assumed that a 1 kW panel produces on average 4.5 kilowatt-hours (kWh) per day in California [73]. The initial cost of the solar panels was found using the levelized cost of energy of solar PV published by US Energy Information Administration [84], which is reported in dollars per MWh. This value was multiplied by the amount of energy produced by the panel in its first year. Hsu et al. also mention that solar PV typically has a performance degradation rate of 0.5% per year, which was used in the performance analysis. Additionally, a lifetime of 25 years was assumed, given that previous studies assumed a lifetime between 20 and 30 years [83,85].

For solar panel installations in parking lots, there are nearly 34,000 parking spots that can be covered by PV panel across Caltrans Ride and Share locations or rest areas. This value includes parking spaces across all Park and Ride locations in the state, as well as rest areas along I-5, I-10, and I-15. The former values were acquired from a 2019 inventory shared by one of the Park and Ride coordinators [86], and is an update to the publicly released 2018 values [87]. The rest area numbers were acquired by manually assessing rest areas along three major interstate highways and counting the existing parking spaces. Since no existing projects were found, this study assesses the installation of solar carports on all parking spaces. The solar canopies are assumed to be wide enough to cover two parking spaces, with support beams placed every three parking spaces. Under this arrangement, the structure provides area to support 48 solar panels that measure 1 by 1.6 meters each. This would provide space for 130,000 meters-squared of PV panel, which results in an installed rated capacity of 18.6 MW. It is further assumed that canopies are, on average, installed in groups of five, such that 30 parking spots (15 long and two wide) are covered by one cohesive solar canopy. The supporting structure is assumed to be all steel, as per the material specifications released by Carport Structures Corporation [88]. The design of the modeled solar canopy was derived from a product bulletin released by Structural Solar [89] and the solar canopy design specifications released by Carport Structures Corporation.

The simplified carport structure model was 9.9 m (32.5 ft) long and 8.2 m (27 ft) wide accommodating six parking spots. It only included the vertical support beams, the lengthwise beams that span the two adjacent parking spaces, and the numerous smaller beams to support the solar panels. One change, however, was to include a cement concrete base that is two and a half feet tall, which is meant to protect the structure from vehicle-related damage; the vertical support beam was shortened accordingly. The structure may need minor repairs after 25 years, but these are considered negligible, and it is therefore assumed that the structure does not need to be replaced until after 2050.

The cost for the solar carports was assumed from the baseline prices listed by Solar Electric Supply Incorporated, a California-based company. Their listed base prices for solar carports with capacities ranging from 50 to 250 kW were between $1.30 and $1.50 per watt. This study used the median value of $1.40 per watt; it is possible that since installation varies by site, true costs could prove to be higher. The time required for installation can vary per site as well. Speaking with a representative from Baja Carports, the typical installation time for 100 spaces is between two and three months. Therefore, it is possible for one installation team to install solar carports over approximately 500 parking spaced per year. Assuming as before that there are 17 such teams across the state, 8500 spaces could be installed per year, and all spaces would be covered by the fourth year.

The carbon intensity of the grid was determined using the grid mix mentioned in the EIA's Annual Energy Outlook, combined with emissions values per fuel source outlined in the GREET 1 model (Argonne National Laboratory, 2017); the value used was 240.36 g $CO_{2\text{-eq}}$ per kWh. Finally, the electricity price used was $0.152 per kWh, which was the average electricity price across all sectors according to a report released by the US EIA [84]. Calculations for profit generation through electricity sales include a 4 percent annual discount rate.

An assessment of the data sources used in our calculation methods is presented in Table A6.

**Table A6.** Data quality assessment.

| Categories | Data Sources | Data Quality | | | | | | | |
|---|---|---|---|---|---|---|---|---|---|
| | | Reliability | Geography | Time | Technology | Completeness | Reproducibility | Representativeness | Uncertainty |
| **Data Type** | | | | | | | | | |
| Annual solar energy generation | Sendy [90] | Fair | US | Good | Very Good | Fair | Yes | Yes | Low |
| Solar PV degradation rate | Hsu et al. [83] | Very Good | US | Fair | Very Good | Very Good | Yes | Yes | Low |
| Annual wind energy generation | Smoucha et al. [74] | Very Good | US | Fair | Very Good | Very Good | Yes | Yes | Low |
| Turbine degradation rate | Staffel and Green [77] | Very Good | US | Good | Very Good | Very Good | Yes | Yes | Low |
| **LCA Related** | | | | | | | | | |
| Wind Turbine | Smoucha et al. [74] | Good | EU | Fair | Very Good | Very Good | Yes | Yes | Low |
| Solar Panel | Hsu et al. [83] | Very Good | US | Fair | Very Good | Very Good | Yes | Yes | Low |
| Electricity | US EIA [84] | Very Good | US | Good | Very Good | Very Good | Yes | Yes | Low |
| Steel | EcoInvent [70] | Good | Global | Fair | Very Good | Fair | Yes | Yes | High |
| Cement Concrete | Saboori et al. [91] | Very Good | US | Very Good | Very Good | Very Good | Yes | Yes | Low |
| **Cost Related** | | | | | | | | | |
| Wind Turbine | Wiser and Bolinger [82] | Very Good | US | Very Good | Very Good | Good | Yes | Yes | Low |
| Solar Panel | US EIA [84] | Good | US | Good | Very Good | Good | Yes | Yes | High |
| Electricity | US EIA [84] | Very Good | US | Good | Very Good | Good | Yes | Yes | Low |
| Steel | Focus Economics | Good | US | Very Good | Very Good | Good | Yes | Yes | Low |
| Solar Carport | Solar Electric Supply Inc. [89] | Very Good | US | Very Good | Very Good | Good | Yes | Yes | Low |

### Table A7. Questionnaire for Strategy 2.

| Question Number | Question | Answer |
|---|---|---|
| 1. | Define change | a. EXISTING: *Caltrans has installed solar panels on buildings.*<br>b. CHANGE: *Install wind turbines and solar panels all physically possible places with a reasonable payback period.* |
| 2. | Define the state of readiness of the change of technology (using approach adapted from NASA) | *Solar canopies over parking spaces: TRL 9: actual system proven in operational environment elsewhere or less-than-full market penetration. Wind turbines in interchanges and solar panel along right-of-ways: TRL 5 and 6: technology validated and demonstrated in relevant environment at less than full scale.* |
| 3. | Define system in which change occurs | *Caltrans owned and operated state highway network and other land/property assets. Cost to be carried within existing budgets unless other funds found, bonds, CAP and Trade, or additional state funding increase in budget. Budget constraint optimization and unconstrained optimization. Cannot be the only criteria for funding.* |
| 4. | Will the market change or is it just changes in market share? | *No* |
| 5. | Who is responsible for change? | *Caltrans. State transport agency, CTC, legislature, energy commission, CPUC* |
| 6. | Who is responsible for implementing change? | *Caltrans* |
| 7. | Who pays for change | a. Government, level of government *State government, passed on to consumers*<br>b. Producers without pass through to consumers *n/a*<br>c. Consumers *n/a* |
| 8. | What will drive change (X) | a. Market<br>b. Market incentives X<br>c. Regulation X<br>d. Legislation X<br>e. Internal Policy X<br>f. Public programs incentivizing change<br>g. Education |
| 9. | What will the change do to these other environmental indicators | LCA will answer<br>i. Air pollution<br>ii. Water pollution<br>iii. Energy use<br>   1. Renewable<br>   2. Non-renewable<br>   3. Renewable energy source used as material<br>   4. Non-renewable energy source used as material<br>iv. Water use<br>v. Use of other natural resources |
| 10. | What are the performance metrics in addition to GHG reduction and cost? | - *Safety changes*<br>- *KWh diff times of the day and diff seasons, aesthetics, noise.* |
| 11. | Supply curve calculation questions: | a. Expected change in GHG output per unit of change in system (LCA).<br>b. Expected maximum units of change in system (LCA).<br>c. Time to reach maximum units of change (reasonable time to be implementable), policy question.<br>d. Expected shape of change rate (dependent on the funding):<br>   i. Linear<br>   ii. Increasing to maximum<br>   iii. Decreasing to maximum<br>   iv. S-shaped *(Expected)*<br>e. Estimated initial cost per unit of change<br>f. Estimated life cycle cost per unit of change (LCCA) |

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
