# Peer review of "Life Cycle Assessment for Transportation Infrastructure Policy Evaluation and Procurement for State and Local Governments"

_sustainability, doi:10.3390/su11226377_

Round 1
Reviewer 1 Report
I thank the authors for their efforts to improving the quality of the manuscript.
1-While I would have liked to see much more concrete effort to situate the work in the policy literature aside from adding two citations from givoni and taegag, and further improvements to sections 1-3 I can live with most of the disclaimers added to the text in regards to scale up and the changes made in the paper with the exception of section 3.
Development of the framework based on the experience without appropriate connection to the literature is not good enough IMHO. I urge the authors to appropriately improve section 3 with better providing rationale for the framework and using literature such as the case of item 2 of the framework which cites references [33]. All of these have to be cited and justified. In addition, how do different aspects of the framework tie together? can you develop a schema to help the readers to see the big picture?
2-I appreciate authors focus on local and state level. Please update the title of the manuscript to reflect this appropriately. It seems that policy evaluation is then only appropriate at the local and state level rather than the national level and cannot be scaled up. Changing "USA" to local/state levels or using California in the title is more appropriate.
Reviewer 2 Report
The topic is of interest to the readers, however, the information could be presented in a more clear way. Page numbers would have been helpful for the review.
The article is very generously written and many of the sentences would be more easily understandable if they would be shorter. For example page 3 lines 116-121 (that forms one sentence) could have been cut into at least three sentences. Other examples can be found throughout the text. However, the language itself is well written.
There is also a lot of information that's relevance to the topic is unclear such as page 2 lines 58-64.
What do the different patterns in Figure 1 present?
The costs for scenario 1 and 2 differ only by 1.3% (Page 13), what is the error margin of the results, since this difference does not seem significant even though the cost difference of 283 Million dollars sounds significant.
On page 18 should be figure 5, not figure 1.
Reviewer 3 Report
Major comments:
The paper presents an approach used at National level for development abatement strategies for GHG reduction,analyzing policy strategies in USA. The structure of the paper and the presented results are clears and well developed. In the introduction, the state of the art could be enriched with actual References, with recent year publication, after to introduce the California case. The conclusions are clear, references style can be improved. Only little modifications need after the publication.
Minor comments:
-Line 22 greenhouse gas (GHG) you have just used this abbreviation in line 21. You can use also only GHGs.
-Line 40. Please, add this References:
-Colvile, R.N, Hutchinson, E.J., Mindell, J.S., Warren, R.F. The transport sector as a source of air pollution (2001) Atmospheric Environment, 35 (9), pp. 1537-1565. doi: 10.1016/S1352-2310(00)00551-.
-Hooftman, N., Oliveira, L., Messagie, M., Coosemans, T., Van Mierlo, J. Environmental analysis of petrol, diesel and electric passenger cars in a Belgian urban setting (2016) Energies, 9 (2), pp. 1-24. doi: 10.3390/en9020084
-Line 41. Please, add this References:
- Marseglia, G. Rivieccio, E. Medaglia, C.M., “The dynamic role of Italian Energy strategies in the worldwide scenario”, Kybernetes, Volume 48, Issue 3, 4 March 2019, pp. 636-649, https://doi.org/10.1108/K-04-2018-0199.
-Shayegh, S., Sanchez, Daniel L., Caldeira, K., 2017. “Evaluating relative benefits of different types of R&D for clean energy technologies”. Energy Policy, Vol. 107, Issue C, pp. 532-538.
- Manni, M. Coccia, V., Nicolini, A., Marseglia, G., Petrozzi, A. “Towards Zero Energy Stadiums: the case study of the Dacia Arena in Udine, Italy”, Energies, 2018, 11, 2396; doi:10.3390/en11092396.
Shayegh, S., Sanchez, Daniel L., Caldeira, K., 2017. “Evaluating relative benefits of different types of R&D for clean energy technologies”. Energy Policy, Vol. 107, Issue C, pp. 532-538.
Sovacool, B.K., Geels, F.W., 2016. “Further reflections on the temporality of energy transitions: a response to critics”. Energy Research & Social Science, Vol.22, pp. 232-237, http://dx.doi.org/10.1016/j.erss.2016.08.013.
-Line 62 :"GHG" you have just used this kind of abbreviation in the abstract. You can use only GHG
-Line 116-121 Please improve english and this sentence
-Line 117 "LCA". You have just defined the first time in Line 27 in the abstract. You can use the abbreviation.
-Lines 159-161 This sentence doesn't add additional value as investigation: it is only the definition of the LCA methodlogy. This sentence could be deleted.
-Line 417 "Pavement Management System" (PMS)
-Line 422 Full Depth Reclamation (FDR)
-LIne 423 Cold In Place Recycling (CIR)
-LIne 465 Passenger Car Equivalent (PCE)
-LIne 508-509: the abbreviation CWZ is defined the first time in line 505
-Line 558 Alternative Fuel Vehicles (AFV)
-Line 792 Figure 5 and not 1. Please correct.
Appendix A and B could be inserted in some paragraph in the text.
-References.Please check these refs:
[37] 2014"."
[44] "Vol.60, 97-131.
[61], [62], [63], [69],[73],[74] pp. and not p.p.
-Where is possible you can also add the doi in the references.
Round 2
Reviewer 1 Report
Paper has improved over the previous draft and is now ready for publication.
This manuscript is a resubmission of an earlier submission. The following is a list of the peer review reports and author responses from that submission.
Round 1
Reviewer 1 Report
please see the attached file.

Reviewer 2 Report
This paper aims to present an alternative methodology to be used for looking at the tradeoff of different mitigation options for climate change and uses 2 case studies to demonstrate the usefulness of the approach for “Optimized Triggering of Pavement Roughness to Reduce GHG and in “Installing solar and wind energy technologies within the state highway network right-of-way”.
The text of the paper is clear and well written and does not need modification.
The authors seem to be categorically unaware of a whole subfield in public policy that is dedicated to addressing the issues they are interested to address in this paper. I highly recommend reading the works of Michael Howlett, Eran Fietelson, Moshe Givoni, Araz Taeihagh, and OECD on how to design policies and examine alternative choices and situate their work within the appropriate literature and focus more on the shortcomings of their approach and whether it can be scaled up beyond the cases the use or is it better to present it as a local level procurement evaluation methodology rather than making claims about policy relevance given that the cases used have not considered the most important aspects considered in policy design and implementation.
Section 3 as it stands is completely useless. Instead of these bullet points please provide the details and rationale for these choices or reference each to appropriate works.
The two case studies presented in the paper are extremely limited and do not demonstrate the possibility for scale up.
In addition, the limitations of the approach as highlighted at the end of each case demonstrate that the usability of the approach for addressing societal issues and policy-relevant outcomes is extremely limited.